# Towards a Mechanistic Explanation of Diffusion Model Generalization

**Matthew Niedoba** [1 2]  **Berend Zwartsenberg** [2]  **Kevin Murphy** [1]  **Frank Wood** [1 2 3]

## Abstract

We propose a simple, training-free mechanism which explains the generalization behaviour of diffusion models. By comparing pre-trained diffusion models to their theoretically optimal empirical counterparts, we identify a shared local inductive bias across a variety of network architectures. From this observation, we hypothesize that network denoisers generalize through localized denoising operations, as these operations approximate the training objective well over much of the training distribution. To validate our hypothesis, we introduce novel denoising algorithms which aggregate local empirical denoisers to replicate network behaviour. Comparing these algorithms to network denoisers across forward and reverse diffusion processes, our approach exhibits consistent visual similarity to neural network outputs, with lower mean squared error than previously proposed methods.

## 1. Introduction

Diffusion models (Sohl-Dickstein et al., 2015; Ho et al., 2020; Song et al., 2021) have become the de facto standard for modelling image (Rombach et al., 2022) and video (Harvey et al., 2022) data due to their high sample quality and generalization abilities. When properly tuned, diffusion models produce samples that are distributionally similar to their training set, but are not exact copies of training data (Zhang et al., 2023).

This behaviour is remarkable, as linear increases in data dimensionality require exponentially more training samples to model the data density (Bellman, 1966). Avoiding this curse of dimensionality requires inductive biases that enable generalization from sparse examples (Goyal & Bengio, 2022). Recently, research has found that diffusion models produce

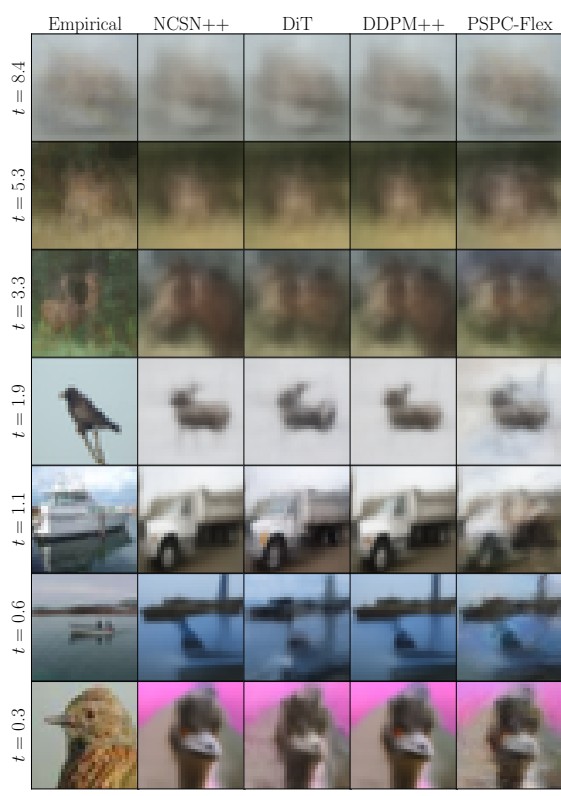

*Figure 1.* Denoiser outputs given shared reverse process noisy inputs from CIFAR-10. **Column 1**: *Optimal* empirical denoiser, i.e. what a "perfect" neural network denoiser would output if appropriately parameterized and trained to achieve minimal loss on the diffusion denoising loss (Equation (5)). **Columns 2-4**: Outputs from various denoising neural networks. For $t < 3.3$ all networks deviate from the optimal denoiser in similar ways. **Column 5**: Our learning-free patch set posterior composite denoiser produces qualitatively similar outputs to the neural network denoisers, suggesting that neural networks may generalize in part via patch denoising and composition.

near-identical samples despite differences in architecture, optimization, or diffusion hyperparameters (Zhang et al., 2023), suggesting the presence of inductive biases common to all image diffusion models.

Diffusion models generate samples through an iterative process in which noise is progressively removed via repeated denoising operations. Crucially, at each step of this process, there exists an *optimal* denoiser function that can be ex-

---

[1]University of British Columbia [2]Inverted AI [3]Alberta Machine Intelligence Institute. Correspondence to: Matthew Niedoba <mniedoba@cs.ubc.ca>.

*Proceedings of the 42$^{nd}$ International Conference on Machine Learning*, Vancouver, Canada. PMLR 267, 2025. Copyright 2025 by the author(s).

pressed as a simple weighted average of the training dataset (Vincent, 2011; Karras et al., 2022). However, using this optimal function in the denoising sampling procedure results in exact replication of the training dataset *without generalization* (Gu et al., 2023). The remarkable generalization capabilities of diffusion models therefore emerge from repeated neural network approximation errors relative to this optimal denoiser. These deviations from optimality, accumulated over the sampling procedure, compound to produce diverse samples.

In this work, we study diffusion model inductive biases through analysis of these network denoiser approximation errors. Using this approach, we find that irrespective of neural network architecture, denoisers make similar approximation errors in both magnitude and quality. Through analysis of the gradients of these denoisers, we find evidence of a shared local inductive bias across image diffusion models.

From this, we hypothesize that neural diffusion model generalization arises in part through locally biased operations. We establish evidence for this hypothesis by approximating these operations with patch-based empirical denoisers. Using these patch estimators, we demonstrate that for large portions of the forward diffusion process, local denoisers are equivalent to regions of the optimal denoiser. Further, we find that over the portion of the sampling procedure in which network denoisers deviate from optimal denoisers, patches of network outputs are closely approximated by patch empirical denoisers.

Finally, we propose our Patch Set Posterior Composites (PSPC) denoiser which aggregates patch empirical denoisers across varying spatial locations to approximate the hypothesized local mechanism of denoiser generalization. Comparing our fully training-free, empirical denoiser to network denoisers, we find PSPC and network denoisers are more similar to each other than to the optimal denoiser (Figure 1). Furthermore, samples produced using our denoiser share structural similarities to those produced by neural-network parameterized diffusion models. These findings provide strong evidence to conclude that patch denoising and composition comprise a significant portion of the generalization behaviour of image diffusion models. Our open-source implementation of PSPC and other denoisers is available at https://github.com/plai-group/pspc.

## 2. Background

Diffusion models are based on a forward diffusion process that gradually adds Gaussian noise to a data distribution $p(\mathbf{x}), \mathbf{x} \in \mathbb{R}^d$. This forward diffusion process can be described through stochastic differential equations of the form

$$d\mathbf{z} = \mathbf{f}(\mathbf{z}, t)dt + g(t)d\mathbf{w}, \tag{1}$$

where $\mathbf{f}(\mathbf{z}, t)$ and $g(t)$ are known as the drift and diffusion functions and $d\mathbf{w}$ is the standard Wiener process (Song et al., 2021).

At every point $t \in (0, T]$, Equation (1) produces marginal latent variable distributions $p_t(\mathbf{z}) = \int p_t(\mathbf{z}|\mathbf{x})p(\mathbf{x})d\mathbf{x}$, $\mathbf{z} \in \mathbb{R}^d$. With appropriate $\mathbf{f}(\mathbf{z}, t)$ and $g(t)$, $p_t(\mathbf{z}|\mathbf{x})$ is a Gaussian distribution with closed form mean and variance. Generally, these $g$ and $\mathbf{f}$ are also selected such that $p_T(\mathbf{z}) \approx \pi(\mathbf{z})$, a simple Gaussian prior.

The aim of diffusion models is to learn the reversal of Equation (1), described by a matching SDE (Song et al., 2021)

$$d\mathbf{z} = \left[ \mathbf{f}(\mathbf{z}, t) - g(t)^2 \nabla_{\mathbf{z}} \log p_t(\mathbf{z}) \right] dt + g(t)d\tilde{\mathbf{w}} \tag{2}$$

Starting from any $p_T(\mathbf{z}) \approx \pi(\mathbf{z})$, every marginal distribution of the solutions to Equation (2) match those of Equation (1). Notably, the reverse time SDE has a corresponding probability flow ODE (PF-ODE) which also shares this property

$$\mathbf{z} = \left[ \mathbf{f}(\mathbf{x}, t) - \frac{1}{2} g(t)^2 \nabla_{\mathbf{z}} \log p_t(\mathbf{z}) \right] dt. \tag{3}$$

Although multiple choices of $\mathbf{f}(\mathbf{z}, t)$ and $g(t)$ are possible, Karras et al. (2022) demonstrate that many such choices are equivalent. We therefore adopt their parameterization with $\mathbf{f}(\mathbf{z}, t) = \mathbf{0}$ and $g(t) = \sqrt{2t}$ resulting in transition distributions $p_t(\mathbf{z} \mid \mathbf{x}) = \mathcal{N}\left(\mathbf{x}, t^2 \mathbf{I}_d\right)$ and prior $\pi(\mathbf{z}) = \mathcal{N}\left(\mathbf{0}, T^2 \mathbf{I}_d\right)$. Note that for these choices, the standard deviation of the added noise is $\sigma(t) = t$.

Solving Equation (2) or Equation (3) requires estimation of $\nabla_{\mathbf{z}} \log p_t(\mathbf{z})$, known as the score function. For our choice of diffusion process, the score has the form

$$\nabla_{\mathbf{z}} \log p_t(\mathbf{z}) = \frac{\mathbb{E}\left[\mathbf{x} \mid \mathbf{z}, t\right] - \mathbf{z}}{t^2}. \tag{4}$$

From Equation (4), score estimation is equivalent to estimating the posterior mean $\mathbb{E}\left[\mathbf{x}|\mathbf{z}, t\right]$, an operation referred to as *denoising*. As the analytic form of $p(\mathbf{x})$ is generally unknown, exact computation of the posterior $p_t(\mathbf{x}|\mathbf{z})$ and therefore $\mathbb{E}\left[\mathbf{x}|\mathbf{z}, t\right]$ is intractable. Instead, diffusion models use neural-network denoisers to approximate $\mathbb{E}\left[\mathbf{x}|\mathbf{z}, t\right]$. These denoisers are trained using an *empirical* data distribution $p_{\mathcal{D}}(\mathbf{x}) = \frac{1}{N} \sum_{\mathbf{x}^{(i)} \in \mathcal{D}} \delta(\mathbf{x} - \mathbf{x}^{(i)})$, with dataset $\mathcal{D} = \left\{\mathbf{x}^{(1)}, \ldots, \mathbf{x}^{(N)} \mid \mathbf{x}^{(i)} \sim p(\mathbf{x})\right\}$, by minimizing

$$\mathbb{E}_{\mathbf{x}^{(i)} \sim p_{\mathcal{D}}(\mathbf{x}), \mathbf{z} \sim p_t(\mathbf{z}|\mathbf{x}^{(i)}), t \sim p(t)} \left[ \lambda(t) \left\| \mathbf{x}^{(i)} - D_\theta(\mathbf{z}, t) \right\|_2^2 \right] \tag{5}$$

where $\lambda(t)$ is a weighting parameter. The minimizer of Equation (5) and *optimal* denoiser for any $(\mathbf{z}, t)$ is the empirical posterior mean (Vincent, 2011; Karras et al., 2019)

$$\mathbb{E}_{\mathbf{x} \sim p_{\mathcal{D}}} \left[\mathbf{x}|\mathbf{z}, t\right] = \sum_{\mathbf{x}^{(i)} \in \mathcal{D}} p_t(\mathbf{x}^{(i)}|\mathbf{z})\mathbf{x}^{(i)} \tag{6}$$

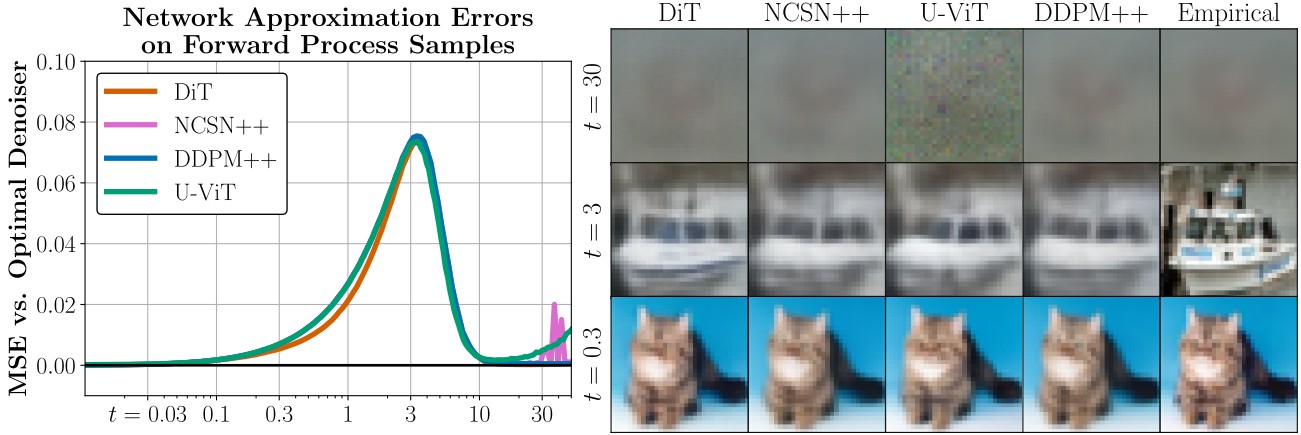

*Figure 2.* **Left**: Mean squared error between empirical and network denoisers for three architectures on CIFAR-10. **Right**: Comparison of network and empirical denoiser for a shared $\mathbf{z} \sim p_t(\mathbf{z}|\mathbf{x})$ at three $t$ values. Network estimators have low error for small and large $t$, but large errors around $t = 3$. At this point, each network varies substantially from the empirical denoiser *in the same way*.

which is a simple average over the images of the dataset $\mathcal{D}$, weighted by their posterior probability

$$p_t(\mathbf{x}^{(i)}|\mathbf{z}) = \frac{p_t\left(\mathbf{z}|\mathbf{x}^{(i)}\right)}{\sum_{\mathbf{x}^{(j)} \in \mathcal{D}} p_t\left(\mathbf{z}|\mathbf{x}^{(j)}\right)}. \quad (7)$$

Hereafter, we refer to the denoiser of Equation (6) as the optimal denoiser.

## 3. Inductive Biases of Network Denoisers

Although Equation (6) is the optimal solution to the diffusion score matching objective, sampling using this denoiser can only reproduce exact copies of training data (Gu et al., 2023). The presence of generalization in image diffusion models beyond their training set therefore implies that denoiser networks make approximation errors relative to the optimal empirical denoiser. Further, the similarity of diffusion samples across several confounding factors (Zhang et al., 2023) suggests a shared class of approximation bias. To understand the generalization of diffusion models, we must understand and characterize these approximation errors.

To begin, we simply compare the denoiser outputs of four unconditional diffusion models trained on CIFAR-10 (Krizhevsky et al., 2009) to the optimal denoiser of that dataset. We evaluate models parameterized by NCSN++ (Song et al., 2021; Karras et al., 2022), DDPM++(Song et al., 2021; Karras et al., 2022), DiT (Peebles & Xie, 2023) and U-ViT (Bao et al., 2022) architectures[1].

Figure 2 plots the mean squared error (MSE) between network and optimal denoisers, evaluated over 150 discrete

[1]For architecture details, see Appendix A.

values of $t \in [0.01, 100]$. For each $t$, we compute MSE over 10,000 $\mathbf{z} \sim p_t(\mathbf{z}|\mathbf{x})p_{\mathcal{D}}(\mathbf{x})$ drawn from the forward process. Across all architectures, we observe similar behaviour to Niedoba et al. (2024), Figure 3 - that network denoisers exhibit low MSE for both small and large values of $t$, but substantial error for $t \in [0.3, 10]$. From the the right portion of Figure 2, only the middle $t = 3$ row has significant differences between the optimal and network denoiser outputs[2]. At this $t$, we observe that all three networks make *qualitatively similar approximation errors*, despite significant differences in architecture and training hyperparameters (see Appendix A). This observation builds upon those of Zhang et al. (2023), suggesting that the similarity of samples produced by generalizing diffusion models is the product of corresponding similarities in denoiser outputs throughout the diffusion process. These denoiser output similarities further imply that denoiser approximation errors are not random optimization artifacts, but the result of a shared inductive bias common to all image network denoisers.

To focus our analysis and avoid redundancy, we hereafter restrict our attention to DDPM++ as a representative network denoiser. Additional quantitative and qualitative results for NCSN++, DiT, and U-ViT denoisers, confirming the consistency of our results across diverse architectures, are provided in Appendix D.

### 3.1. Network Denoiser Gradients

One potential inductive bias (Goyal & Bengio, 2022) of network denoisers is local inductive bias, where denoiser outputs are more sensitive to spatially local perturbations of

[2]The noise in U-ViT's output for $t = 30$ is due to the amplification problem for $\epsilon$-predictor networks (Karras et al., 2022).

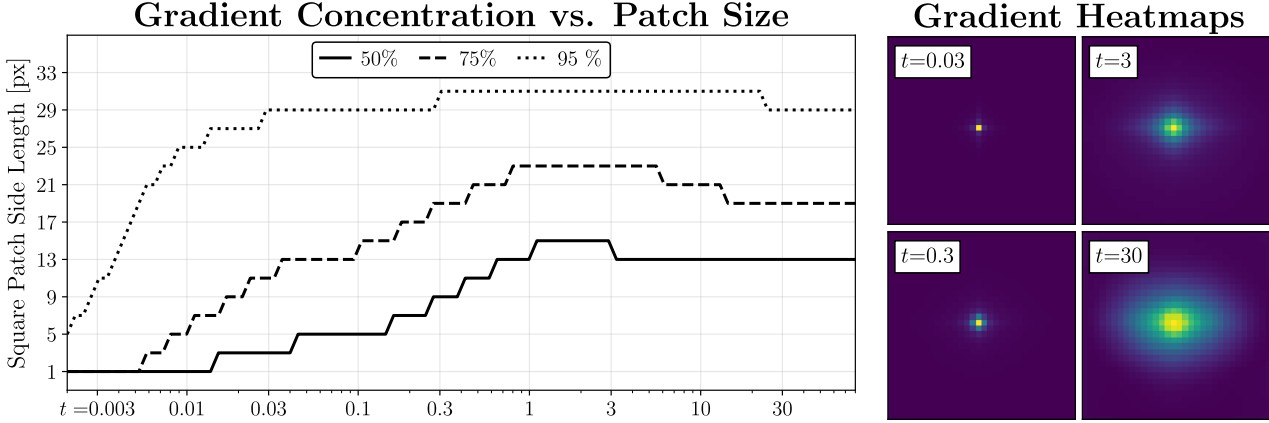

*Figure 3.* **Left**: Comparison of average square patch sizes required to capture 50%, 75%, and 95% of the total gradient sensitivity heatmap. As $t$ increases, larger patch sizes are required to capture a fixed percentage of the total gradient sensitivity heatmap. **Right**: Gradient sensitivity heatmaps for DDPM++ denoiser on CIFAR-10 for output pixel (15,15) across varying $t$.

$\mathbf{z}$ than distant ones. We investigate potential local inductive bias in network denoisers by measuring the sensitivity of the DDPM++ denoiser (Karras et al., 2019) to each input pixel through their gradients. For each $t$, we define the gradient sensitivity heatmap as

$$\mathbf{G}(x, y, t) = \mathop{\mathbb{E}}_{\mathbf{z} \sim p_t(\mathbf{x}^{(i)}, \mathbf{z})} \left[ \sum_{c=1}^{3} |\nabla_{\mathbf{z}_c} D_\theta(\mathbf{z}, t)_{x,y,c}| \right]. \quad (8)$$

Here, $D_\theta(\mathbf{z}, t)_{x,y,c}$ indicates the output of the network denoiser at spatial position $(x, y)$ and channel $c$ where where $x \in 1, \ldots, w, y \in \{1, \ldots, h\}$, and $c \in \{1, 2, 3\}$ for an RGB image of height $h$ and width $w$.

$\mathbf{G}(x, y, t)$ captures the channel-averaged absolute gradient of the network denoiser output at pixel $(x, y)$ with respect to $\mathbf{z}$, a measure of the sensitivity of an output pixel of the network denoiser to each input pixel. In practice, we evaluate the expectation of Equation (8) using 1,000 $\mathbf{z}$ samples drawn from the forward process per $t$. We plot four such heatmaps in Figure 3.

The right portion of Figure 3 confirms that network denoisers demonstrate strong local inductive bias. At $t = 0.03$, the network denoiser is almost exclusively sensitive to the same spatial location as the output pixel. As $t$ increases, so does the area over which the input gradient is concentrated.

We quantify this local inductive bias by measuring the average side length of a square patch centered at pixel $(x, y)$ required to capture a fixed percentage of $\sum_{i,j} \mathbf{G}(x, y, t)_{i,j}$. We plot this quantity in the left panel of Figure 3. For all $t$, the majority of the input gradient is concentrated within a square $15 \times 15$ pixel region around the output pixel. Further, the strength of the local inductive bias is inversely correlated

with $t$. While a $13 \times 13$ patch is required to capture 50% of $\sum_{i,j} \mathbf{G}(x, y, t)_{i,j}$ at $t = 30$, the same percentage can be accounted for in a $3 \times 3$ patch at $t = 0.03$.

Figure 3 provides preliminary evidence that network denoisers are predominantly *locally* sensitive to changes in $\mathbf{z}$. This is somewhat surprising, as the optimal denoiser is by definition *globally* sensitive to any changes in $\mathbf{z}$. Since $p_t(\mathbf{x}^{(i)}|\mathbf{z}) \propto p_t(\mathbf{z}|\mathbf{x}^{(i)})$ and $p_t(\mathbf{z}|\mathbf{x}^{(i)})$ is Gaussian, the posterior probability of any $\mathbf{x}^{(i)}$ is related to the squared distance between every pair of pixels in $\mathbf{x}^{(i)}$ and $\mathbf{z}$.

## 4. Local Denoising Mechanisms

Section 3 presents strong evidence that diffusion model denoisers deviate substantially and similarly from the optimal denoiser. Moreover, Section 3.1 finds that network denoiser gradients show evidence of local inductive bias. However, Figure 2 also shows that network denoisers accurately estimate the empirical posterior mean for most $t$, despite the inherent global sensitivity of this quantity. To resolve this apparent contradiction, we hypothesize that network denoisers perform local computations whose combined result is equivalent to the optimal denoiser for most values of $t$.

One example of such a local computation is denoising over patches of the input $\mathbf{z}$. Formally, we denote cropping matrices as $\mathbf{C} \in \{0, 1\}^{n \times d}$. Then, for any such $\mathbf{C}$, which produces patches $\mathbf{x}_{\mathbf{C}}^{(i)} = \mathbf{C}\mathbf{x}^{(i)}$ and $\mathbf{z}_{\mathbf{C}} = \mathbf{C}\mathbf{z}$, we define the patch posterior with $p_t(\mathbf{z}_{\mathbf{C}} \mid \mathbf{x}_{\mathbf{C}}^{(i)}) = \mathcal{N}(\mathbf{x}_{\mathbf{C}}^{(i)}, t^2 \mathbf{I}_n)$ as

$$p_t(\mathbf{x}_{\mathbf{C}}^{(i)} \mid \mathbf{z}_{\mathbf{C}}) = \frac{p_t\left(\mathbf{z}_{\mathbf{C}} \mid \mathbf{x}_{\mathbf{C}}^{(i)}\right)}{\sum_{\mathbf{x}^{(j)} \in \mathcal{D}} p_t\left(\mathbf{z}_{\mathbf{C}} \mid \mathbf{x}_{\mathbf{C}}^{(j)}\right)} \quad (9)$$

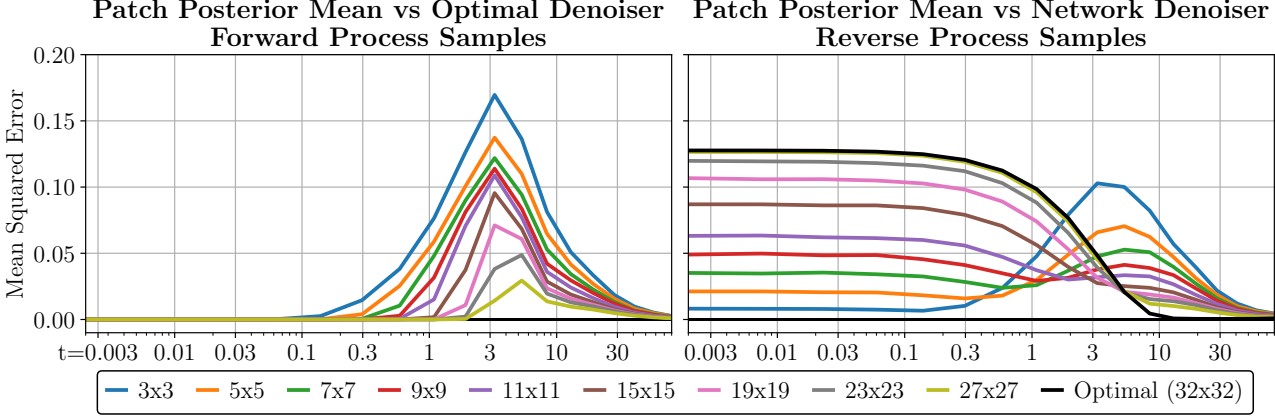

Figure 4. **Left**: Comparison of patch posterior means with varying patch sizes to corresponding patches of the optimal denoiser over forward process samples. For $t < 1$, relatively small square patch posterior means *exactly* match the optimal denoiser. As $t$ increases, larger patch sizes are required to exactly estimate patches of the optimal denoiser. **Right**: Comparison of patch posterior means with varying patch sizes to DDPM++ denoiser patches on $\mathbf{z}$ drawn from the reverse process. Patch posterior means estimate the network denoiser patches better than the optimal denoiser for all $t < 3$. As $t$ decreases, so does the patch size which best estimates network denoiser patches.

and the empirical patch posterior mean as

$$\mathbb{E}_{p_{\mathcal{D}}}\left[\mathbf{x}_{\mathbf{C}} \mid \mathbf{z}_{\mathbf{C}}, t\right] = \sum_{\mathbf{x}^{(i)} \in \mathcal{D}} p_t\left(\mathbf{x}_{\mathbf{C}}^{(i)} \mid \mathbf{z}_{\mathbf{C}}\right)\mathbf{x}_{\mathbf{C}}^{(i)}. \quad (10)$$

Like the empirical denoiser of Equation (6), the patch posterior mean is a simple average over dataset elements, weighted by posterior probabilities. However, unlike the empirical denoiser which uses full images $\mathbf{x}^{(i)}$, the patch posterior mean is computed over a spatial subset of the dataset where each image is cropped by $\mathbf{C}$.

It is particularly convenient to work with the set of square cropping matrices. We define $\mathbf{C}(x, y, s)$ as a square cropping matrix such that $\mathbf{C}(x, y, s)\mathbf{x}^{(i)}$ is the $s \times s$ pixel patch of $\mathbf{x}^{(i)}$ with upper left corner at pixel $(x, y)$. For a patch size $s$ and a square image of spatial size $h \times h$, we denote $\mathcal{C}_s = \{\mathbf{C}(x, y, s) \mid x\{0, \ldots, h - s\}, y \in \{0, \ldots, h - s\}\}$ as the set of all such cropping matrices. Notably, $\mathcal{C}_s$ does not include cropping matrices which would require any padding of $\mathbf{x}^{(i)}$ at the image boundaries.

We have hypothesized that local inductive biases in network denoisers are the result of local denoising operations which approximate the optimal denoiser. However, in general, $\mathbb{E}_{p_{\mathcal{D}}}[\mathbf{x}_{\mathbf{C}} \mid \mathbf{z}_{\mathbf{C}}, t] \neq \mathbf{C}\mathbb{E}_{p_{\mathcal{D}}}[\mathbf{x} \mid \mathbf{z}, t]$ because $p_t(\mathbf{x}_{\mathbf{C}}^{(i)} \mid \mathbf{z}_{\mathbf{C}}) \neq p_t(\mathbf{x}^{(i)} \mid \mathbf{z})$. Why then would we expect network denoisers to use local posterior mean estimates to estimate the global posterior mean?

Critically, there are two cases when these distributions are similar. For sufficiently small $t$ and $\mathbf{z}$ drawn from the forward process, $p_t(\mathbf{x}_{\mathbf{C}}^{(i)} \mid \mathbf{z}_{\mathbf{C}}) \approx p_t(\mathbf{x}^{(i)}|\mathbf{z}) \approx \delta(\mathbf{x}^{(i)})$. Sim-

ilarly, as $t$ becomes large, both posteriors will approach a uniform distribution over $\mathcal{D}$. We note that these two cases correspond to the regions of Figure 2 in which network estimators accurately estimate the empirical posterior mean.

The left portion of Figure 4 empirically confirms these cases of similarity. It plots MSE between $\mathbb{E}[\mathbf{x}_{\mathbf{C}}^{(i)} \mid \mathbf{z}_{\mathbf{C}}, t]$ and $\mathbf{C}\mathbb{E}[\mathbf{x}^{(i)} \mid \mathbf{z}, t]$, averaged across $\mathbf{C} \in \mathcal{C}_s$ and 10,000 $\mathbf{z} \sim p_t(\mathbf{z}, \mathbf{x}^{(i)})$ for varying $t$ and patch sizes $s$. For both $t < 0.1$ and large $t$, patch posterior means are similar to optimal denoiser patches, regardless of patch size. Further, as $t$ increases, larger patch sizes are required to accurately estimate the optimal denoiser. This matches the correlation between local sensitivity and $t$ observed in Figure 3. Notably, the region in which patch posterior means are poor estimators of the optimal denoiser is similar to the regions of Figure 2 in which network denoisers do not match the optimal denoiser.

If network denoisers utilize localized denoising mechanisms to approximate the optimal denoiser over the forward process, we would expect this mechanism to be used for reverse process samples as well. Although patch posterior means approximate optimal denoiser patches well over the forward process, from the right subplot of Figure 4 this is not the case for $\mathbf{z}$ sampled from the reverse process. Instead, we find patch-based denoisers estimate *network denoiser* patches well for all $t < 5$ and especially for $3 \times 3$ patch denoisers when $t < 0.3$. This provides further evidence that network denoisers may utilize local denoising operations such as patch posterior means to approximate the optimal denoiser.

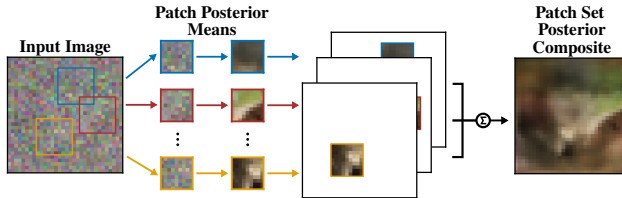

*Figure 5.* Our PSPC denoiser. First, **z** is decomposed into patches using a set of cropping matrices. For each patch, we compute the patch posterior mean via Equation (9). Resulting means are then combined into one image and normalized by the by the number of patches that overlap each pixel. Although square patches are visualized, PSPC can be used with any set of cropping matrices.

# 5. Patch Set Posterior Composites

Figure 4 demonstrates that patch posterior means of varying sizes approximate patches of diffusion network denoisers well over both forward and reverse processes. This finding motivates our proposed methodology, which combines patch posterior means at multiple spatial locations.

Formally, given an arbitrary set of cropping matrices $\mathcal{C} = \{\mathbf{C}_1, \dots, \mathbf{C}_L\}$ which denote these spatial locations, we define our Patch Set Posterior Composite (PSPC) method as

$$D\left(\mathbf{z}, t, \mathcal{C}\right) = \left( \sum_{\mathbf{C} \in \mathcal{C}} \mathbf{C}^\top \mathbf{C} \right)^{-1} \sum_{\mathbf{C} \in \mathcal{C}} \mathbf{C}^\top \mathbb{E}\left[ \mathbf{x}_{\mathbf{C}}^{(i)} \mid \mathbf{z}_{\mathbf{C}}, t \right]. \tag{11}$$

The patch set posterior composite of Equation (11) estimates the patch posterior mean for every patch $\mathbf{C} \in \mathcal{C}$ of **z**. The output is produced by summing each of these patch posterior means together before normalizing by $\left( \sum \mathbf{C}^\top \mathbf{C} \right)^{-1}$, the number of patches which overlap each pixel. This process is described in Figure 5. In practice, we estimate patch posterior means using the fast nearest-neighbour score estimators of Niedoba et al. (2024).

The performance of Equation (11) is dependent on the choice of patch set $\mathcal{C}$. We introduce two possible choices of patch set below which represent two variants of our methodology.

## 5.1. PSPC-Square

A natural choice of patch set is $\mathcal{C}s$, the set of overlapping square patches of spatial size $s \times s$ defined in Section 4. However, Figure 4 shows that the optimal patch size varies substantially across the reverse diffusion process. To adapt our method to this observation, we define a patch size schedule $s(t) : \mathbb{R}^+ \to \{1, \dots, h\}$. The square Patch Set Posterior Composite (**PSPC-Square**) is defined by Equation (9) with $\mathcal{C} = \mathcal{C}_{s(t)}$. In practice, we set s(t) according to Figure 4, choosing patch size at each $t$ with the lowest error versus

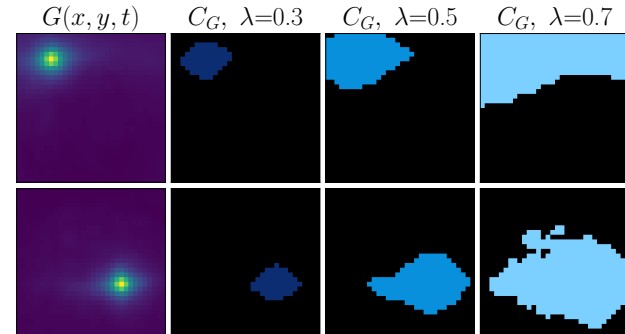

*Figure 6.* CIFAR-10 gradient sensitivity maps and corresponding PSPC-Flex cropping matrices for varying $\lambda$ at $t = 3$. Coloured regions indicate areas cropped by $C_G$. Unlike PSPC-Square, PSPC-Flex patches are adaptive to average network sensitivity.

the network denoiser. Patch size schedules can be found in Appendix B.

## 5.2. PSPC-Flex

While square patches mimic the square receptive fields of convolutional U-Nets, Figure 3 suggests that denoisers gradients are not perfectly square. To enable more flexible, non-square patches of varying shapes and sizes, we introduce **PSPC-Flex**, which utilizes adaptive patch sets based on average gradient sensitivity maps.

We precompute sensitivity maps using Equation (8), averaging DDPM++ denoiser gradients over 1000 forward process **z** samples for each $t$ from the EDM sampling schedule. Then, for each $\mathbf{G}(x, y, t)$, we construct a flexible cropping matrix $\mathbf{C_G}(x, y, t, \lambda)$ by greedily selecting pixels at position $(i, j)$ in descending order of $\mathbf{G}(x, y, t)_{i,j}$ until the cropped region defined by $\mathbf{C_G}(x, y, t, \lambda)$ contains a fixed portion $\lambda \in [0, 1]$ of the total gradient

$$\sum_{i,j} \left( \mathbf{C_G}(x, y, t, \lambda) \mathbf{G}(x, y, t) \right)_{i,j} = \lambda \sum_{i,j} \mathbf{G}(x, y, t)_{i,j}. \tag{12}$$

Figure 6 demonstrates how adaptive cropping matrices better capture the average sensitivity shown in the gradient sensitivity maps. We construct the patch set of PSPC-Flex using these adaptive patches $\mathcal{C}_{\mathbf{G}(t, \lambda)} = \{\mathbf{C_G}(x, y, t, \lambda) \mid x \in \{0, \dots, w - 1\}, y \in \{0, \dots, h - 1\}\}$. Similarly to PSPC-Square, we utilize a time varying threshold function $\lambda(t)$ which is described in Appendix B.

# 6. Results

We evaluate our method on three image datasets – CIFAR-10 (Krizhevsky et al., 2009), FFHQ 64×64 (Karras et al., 2019), and AFHQv2 64×64 (Choi et al., 2020). For each

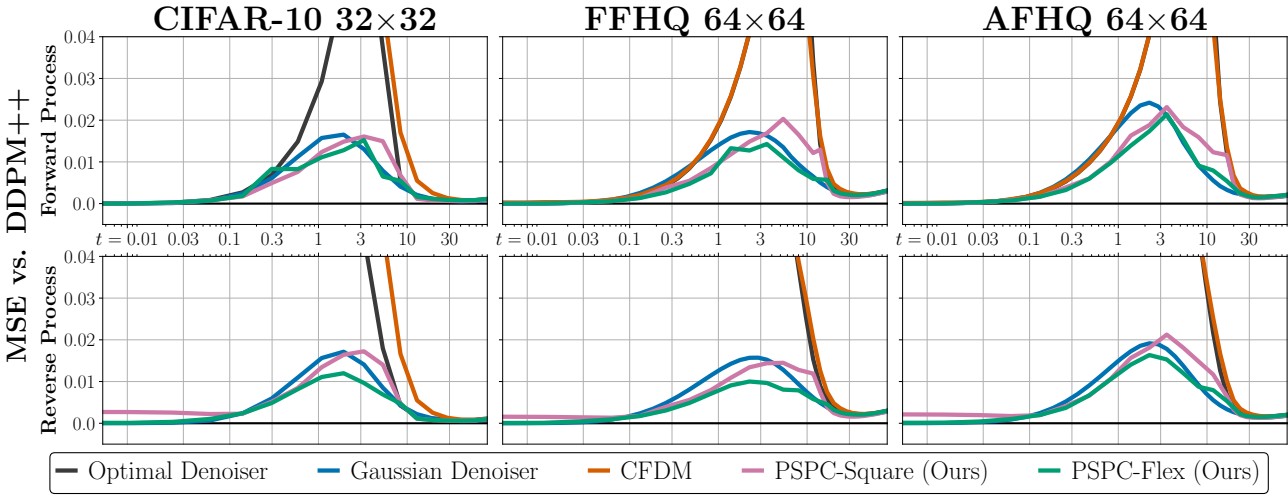

*Figure 7.* Comparison of various denoisers against DDPM++ over forward and reverse processes. For every dataset, PSPC denoisers consistently provide better estimates of DDPM++ outputs than the optimal denoiser. In most cases, PSPC-Flex yields the best estimate of network denoisers, outperforming the Gaussian and CFDM denoisers. For comparisons against other network baselines, see Appendix D.

dataset, we generate an evaluation set of samples drawn from the forward and reverse processes. For both sets, we use the $t$ values defined by the EDM sampling procedure (Karras et al., 2022), with 18 steps for EDM and 40 steps for both FFHQ and AFHQ. To produce our reverse process evaluation set, we utilize pretrained DDPM++ EDM models with their default Heun sampler to generate solutions to Equation (3). For each $t$, we draw $10,000$ z from each process for CIFAR-10 and 2000 **z** for FFHQ and AFHQ.

### 6.1. Network Denoiser Comparison

To evaluate the similarity of PSPC to network denoisers, we compare various empirical denoisers against the EDM DDPM++ denoiser outputs for each dataset. In addition to our PSPC-Square an PSPC-Flex methods, we evaluate the empirical denoiser of Equation (7), the Gaussian denoiser decribed by Wang & Vastola (2024); Li et al. (2024), and the Closed-Form Diffusion Model (CFDM) denoiser of Scarvelis et al. (2023).

Figure 7 plots the MSE of each of the denoisers against the neural network denoiser. We find that for both forward and reverse processes, both PSPC variants have substantially lower MSE against the network denoiser outputs as compared to the empirical and CFDM denoisers. Compared to the Gaussian denoiser, our methods generally have better MSE for $t \in [0.3, 3]$. Comparing our methods, the flexibility of the PSPC-Flex patch set produces better estimates than those of PSPC-Square. Over the reverse process evaluation sets on every dataset, PSPC-Flex generally estimates the network denoiser better than PSPC-Square and all other methods.

Qualitatively, we compare the output of PSPC-Flex to several CIFAR-10 network denoisers and the empirical denoiser in Figure 1. For every $t$, PSPC-Flex outputs are visually similar to the outputs of the various network denoisers, suggesting that local denoising operations comprise a significant portion of the generalization mechanism of diffusion models. Additional denoiser outputs can be found in Appendix E.

### 6.2. PSPC Sampling

Inspired by the remarkable similarity between network denoisers and PSPC-Flex evidenced by Figures 1 and 7, we investigate the efficacy of PSPC-Flex as fully training-free, completely non-neural diffusion model.

Figure 8 compares PF-ODE sampling trajectories using DDPM++ and PSPC-Flex denoisers, starting from a shared $\mathbf{z} \sim \pi(\mathbf{z})$ for each row. PSPC-Flex samples are remarkably similar in structure to those of the diffusion model. For example, the FFHQ samples for both denoisers resemble a brown-haired subject on a blue background. However, the sample quality of PSPC-Flex is consistently worse than those of DDPM++. Although the outputs of both denoisers is generally quite similar at each point of every trajectory, the errors made by PSPC-Flex starting in the middle of each trajectories appear to compound negatively. This leads to substantial visual artifacts in the final samples of PSPC-Flex.

Despite these artifacts, Figure 9 quantitatively demonstrates that PSPC-Flex samples are still significantly more similar to network denoiser samples than the samples produced by the optimal, CFDM, or Gaussian denoisers. We measure similarity using the cosine similarity of SSCD descriptors

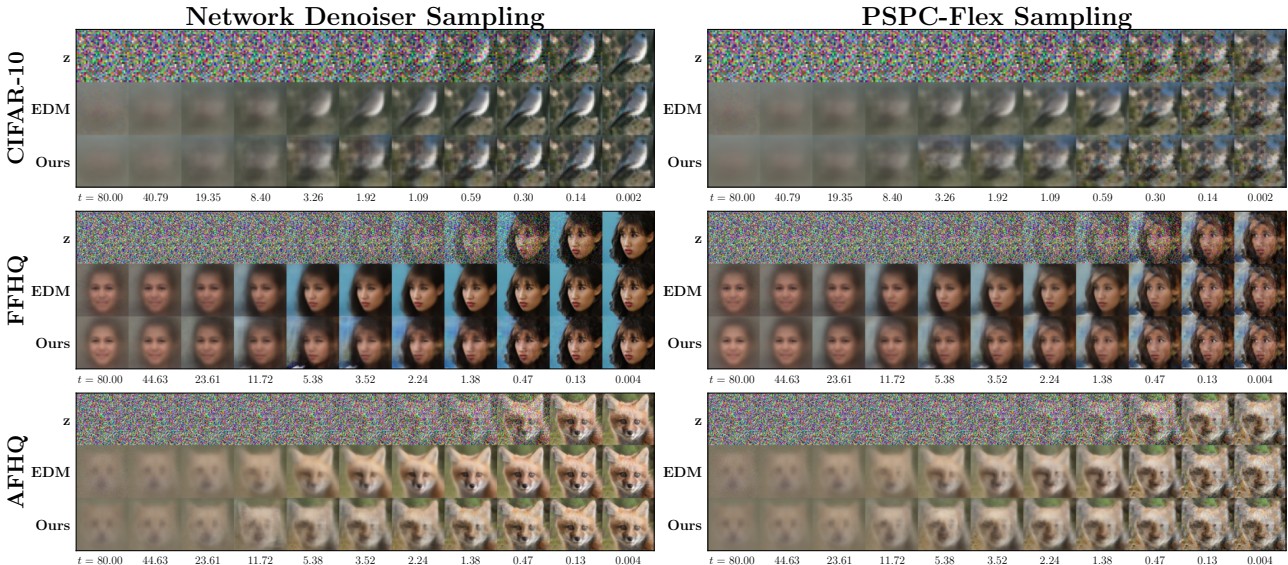

*Figure 8.* PF-ODE sampling trajectories starting from the same initial **z**, comparing the DDPM++ score estimator (Left) with our PSPC-Flex estimator (Right). At each $t$, we display the noisy image **z** along with the output of the DDPM++ denoiser and PSPC-Square on that **z**. In all cases, minor differences between DDPM++ and PSPC-Square outputs only occur over intermediate $t$. Sampling using PSPC-Flex yields samples which are similar in content to the network samples. However, compounding errors lead to suboptimal final samples.

(Pizzi et al., 2022), a embedding model used to detect image copies (details in Appendix C.1). Notably, across all network architectures, PSPC-Flex similarity scores approach 0.6, the threshold used to determine copying by Zhang et al. (2023).

## 7. Related Work

**Diffusion generalization** There is a broad literature considering diffusion generalization. Comparing a variety of networks, Zhang et al. (2023) finds diffusion models produce consistent samples despite differences in architecture and training. Both Niedoba et al. (2024) and Xu et al. (2023) find that diffusion models only deviate from empirical denoisers for intermediate $t$. Further, Niedoba et al. (2024) identify that errors in this region are primarily responsible for diffusion generalization. Similarly, Yi et al. (2023) attributes diffusion generalization to "slight differences" between the network and empirical denoisers. Kadkhodaie et al. (2024) suggest that generalization stems from geometrically adaptive harmonic bases. However, they do not consider how trained models deviate from empirical denoisers.

Several other methods have been proposed to reproduce diffusion generalization. Both Wang & Vastola (2024) and Li et al. (2024) find strong correlations between network denoisers and optimal denoisers under the simplistic model of a Gaussian $p(\mathbf{x})$. Closed-form diffusion models (Scarvelis

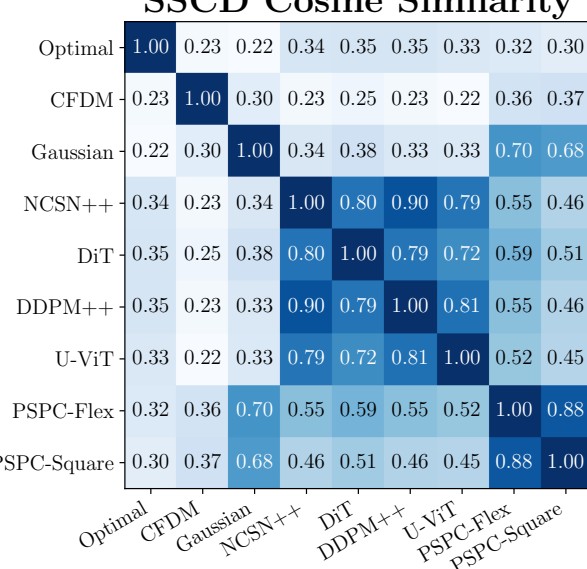

*Figure 9.* SSCD cosine similarity of CIFAR-10 PF-ODE samples produced with varying denoisers from shared **z** initial conditions. Average PSPC-Flex similarity vs. neural-networks ($\mu = 0.55$) is significantly higher than the similarity of optimal ($\mu = 0.34$), CFDM ($\mu = 0.23$), or Gaussian ($\mu = 0.34$) denoisers.

et al., 2023) bias the empirical denoiser with a spectral bias (Rahaman et al., 2019) to produce a generalization mechanism.

Concurrent to our work, Kamb & Ganguli (2024) propose a empirical, patch-based denoiser which leverages empirical patch posterior means and an increasing schedule of square patch sizes to approximate network denoiser outputs. While their method shares similarities with PSPC-Square, our work differs in several key ways. First, PSPC estimates patch-posterior means exclusively over *localized* patch sets. We found this to be particularly beneficial on datasets such as FFHQ, where image features are correlated to specific image positions. Second, while Kamb & Ganguli (2024) focus their analysis on convolutional diffusion denoisers, our work extends to both U-Net and DiT architectures. This broader evaluation is essential to our conclusion that local inductive biases form an important component of diffusion model generalization, regardless of architecture.

**Patch-based Methods** Patch-based methods have long been used in image processing. Classical image denoising methods such as Field of Experts (Roth & Black, 2005) model images as Markov Random Fields to learn patch-based priors. Methods such as expected patch log likelihood (Zoran & Weiss, 2011) and half quadratic splitting (Elad & Aharon, 2006; Zoran & Weiss, 2011; Friedman & Weiss, 2021) leverage such patch-based priors to enable maximum a priori image reconstructions. Importantly, these methods typically focus on recovering clean images rather than estimating posterior means, and typically operate on lower levels of additive Gaussian noise than is typical in the diffusion setting.

Patch-based methods have also been used in image generation. classical single image generation methods synthesize images by ensuring similarity between patches of the input and output images (De Bonet, 1997; Efros & Leung, 1999; Barnes et al., 2009; Simakov et al., 2008) while more recent, deep learning approaches leverage patches to train GANs(Shaham et al., 2019) and diffusion models (Nikankin et al., 2023; Wang et al., 2025). In the area of unconditional image generation, Ding et al. (2023) and Wang et al. (2023) utilize patch-based approaches to improve diffusion model training and performance.

## 8. Conclusions

Our work investigates the generalization mechanisms of image diffusion models. By comparing network denoisers and the optimal denoiser, we find strong evidence that a persistent local inductive bias in network denoisers results in denoiser approximation errors which are both qualitatively and quantitatively similar across all evaluated networks. We hypothesize that this local bias is the result of network denoisers employing local denoising operations to partially

approximate the optimal denoiser. Approximating such local operations with patch posterior means, we find that these means are excellent approximators of optimal denoiser patches over the majority of the forward diffusion process.

By spatially aggregating local patch denoisers, we find the resulting PSPC denoisers are remarkably similar to network denoisers when evaluated at identical $(\mathbf{z}, t)$ inputs. Additionally, PSPC samples share structural similarity to those produced by diffusion models when sampling is identically initialized. We believe that the performance of PSPC is strong empirical evidence that a significant portion of the generalization behaviour of image diffusion models arises from a mechanism which is substantially similar to simple time-varying patch denoising and compositing operations.

Our work has numerous applications. Understanding the mechanisms of diffusion generalization helps to determine the cases when models fail to generalize, mitigating claims of "digital forgery" (Somepalli et al., 2023). Specifically, patch posterior probabilities (Equation (9)) are a promising signal for identifying specific training images which influence the generation of a sample, with copyright and training-data licensing implications. In addition, patch-based diffusion generalization mechanisms may be exploited to improve training and sampling efficiency, as demonstrated by the preliminary exploration of Wang et al. (2023). Finally, further improvements to empirical denoisers such as PSPC-Flex may result in fully training-free models of similar quality to neural diffusion models, eliminating the fiscal and environmental costs of diffusion training.

## Acknowledgements

We acknowledge the support of the Natural Sciences and Engineering Research Council of Canada (NSERC), the Canada CIFAR AI Chairs Program, Inverted AI, MITACS, and Google. This research was enabled in part by technical support and computational resources provided by the Digital Research Alliance of Canada Compute Canada (alliancecan.ca), the Advanced Research Computing at the University of British Columbia (arc.ubc.ca), and Amazon.

## Impact Statement

This paper presents work whose goal is to advance the field of Machine Learning. There are many potential societal consequences of our work, which we discuss here. As generative models become available and are used by the general public, understanding the generalization behaviours is more important than ever. Sentiment towards generative image models has been negatively influenced by instances of dataset reproduction in open-source models. Understanding the mechanisms by which these models generalize is a key stepping stone to preventing such copies. Further, ex-

plicit mechanisms such as those presented in our work may serve as invaluable tools in assigning attribution to generated samples to the source images which inspired them.

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

# A. Network Denoiser Architecture Details

## A.1. CIFAR-10

Our work analyzes diffusion generalization across a diverse set of network architectures, A summary of some of the differences between the networks compared in Figures 2 and 9 is given in Table 1

*Table 1.* Differences in hyperparameters between evaluated CIFAR-10 Network Denoisers

|  | DDPM++ | NCSN++ | DiT | U-ViT |
|---|---|---|---|---|
| Network Architecture | U-Net | U-Net | Vision Transformer | Vision Transformer |
| Output | EDM Residual | EDM Residual | EDM Residual | $\epsilon$ prediction |
| Diffusion Process | EDM | EDM | EDM | Variance Preserving |
| Patch Size | N/A | N/A | 4 | 2 |
| Hidden Size | varies | varies | 768 | 384 |
| Noise Encoding | positional | fourier | positional | token |

For the NSCN++ and DDPM++ architectures, we utilize the pretrained unconditional model checkpoints provided by (Karras et al., 2022). For the Diffusion Transformer, we adopt the code of (Peebles & Xie, 2023), utilizing the EDM preconditioning scheme and data augmentation pipeline. We trained a DiT-B/4 network from scratch on 200 million examples using the Adam (Kingma & Ba, 2015) optimizer and the hyperparamters given in Table 2

*Table 2.* Hyperparameters for DiT training on CIFAR-10

| Hyperparameter | Value |
|---|---|
| Batch Size | 512 |
| Learning Rate | 0.0001 |
| $\beta_1$ | 0.9 |
| $\beta_2$ | 0.999 |
| $\epsilon$ | 1E-8 |
| Patch Size | 4 |
| # Heads | 12 |
| Hidden Size | 768 |
| Transformer Blocks | 12 |
| Dropout Ratio | 0.12 |
| Augmentation Rate | 0.12 |

For U-ViT (Bao et al., 2022), we utilize their pretrained checkpoint. Since U-ViT is an $\epsilon$ prediction network, with a variance preserving diffusion process, we convert $\epsilon$ predictions from their network into $\mathbf{x}$ predictions through the equation

$$\mathbf{x} = \frac{\mathbf{z} - t(t)\epsilon}{s(t)} = \mathbf{x} \tag{13}$$

In general, the diffusion models analyzed in this work produce high quality samples, as measured by Frechet Inception Distance (FID) (Heusel et al., 2017). The 50k FID scores for each model are given below in Table 3

*Table 3.* FID Scores for CIFAR-10 diffusion models

| Model | FID |
|---|---|
| DDPM++ (Karras et al., 2022) | 1.97 |
| NCSN++ (Karras et al., 2022) | 1.98 |
| DiT++ (Peebles & Xie, 2023) | 9.08 |
| U-ViT (Bao et al., 2022) | 3.08 |

## A.2. FFHQ and AFHQ

For all FFHQ and AFHQ experiments, we utilize a pretrained DDPM++ EDM checkpoint (Karras et al., 2022).

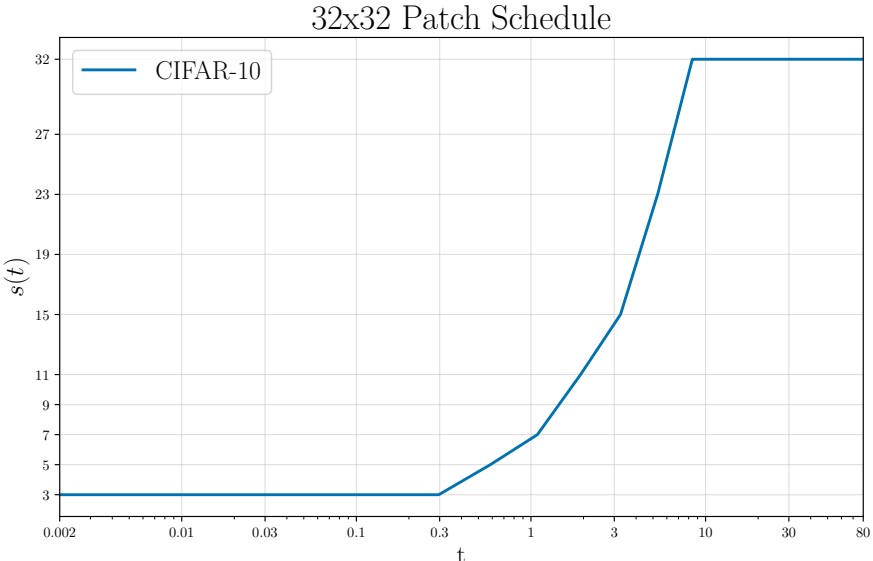

*Figure 10.* Patch Schedule for 32x32 Datasets

## B. Patch Composite Algorithm Details

For each dataset, we tuned an individual schedule for the patch size and gradient threshold for PSPC-Square and PSPC-Flex respectively. Schedules were tuned by minimizing the average path posterior mean error over forward process samples for each dataset. Figures 10 and 11 shows the patch schedule $s(t)$ used for all PSPC-Square results while Figure 12 shows the threshold schedule for all PSPC-Flex results.

## C. Experimental Details

### C.1. SSCD Details

To compute the SSCD cosine simiilarities of Figure 9, we use the `sscd_imagenet_mixup` checkpoint provided by the official SSCD github repository (Pizzi et al., 2022). SSCD is a self-supervised method trained which produces a image descriptor which is similar among copied images.

For our application, we started from a shared set of 1000 $\mathbf{z} \sim \pi(\mathbf{z})$ and then used each of the denoisers listed in Figure 9 to draw samples from those initial points. We used deterministic PF-ODE sampling with an Euler solver and the EDM sampling schedule (Karras et al., 2022). We then encoded each denoiser's samples using the SSCD encoder and computed cosine similarities between each set of images

$$D_{\text{SSCD}}(\mathbf{x}_1, \mathbf{x}_2) = \frac{\texttt{sscd}(\mathbf{x}_1) \cdot \texttt{sscd}(\mathbf{x}_2)}{\|\texttt{sscd}(\mathbf{x}_1)\| \|\texttt{sscd}(\mathbf{x}_2)\|} \tag{14}$$

We averaged the cosine similarities across the 1000 images per denoiser to obtain the results in Figure 9.

## D. Additional Network Baselines

Figure 2 compares four network denoisers on CIFAR-10 and finds that their outputs are qualitatively similar, and that they have similar mean squared error when compared against the optimal denoiser. Due to the similarity between networks, for clarity of presenation we utilized DDPM++ in Figures 3, 4 and 7 and did not present additional results using NCSN++, DiT or U-ViT network denoisers. For completeness, we have included additional versions of these figures below

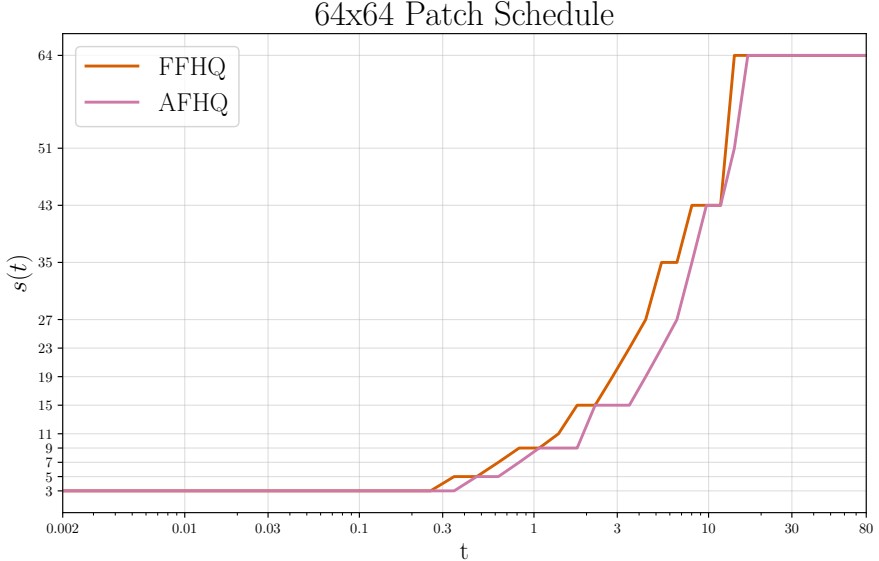

*Figure 11.* Patch Schedules for 64x64 Datasets

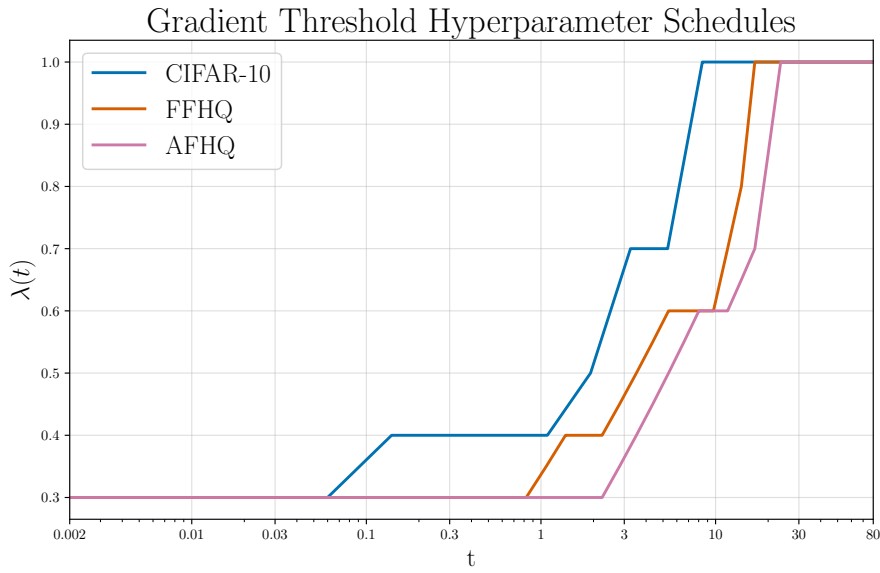

*Figure 12.* Gradient threshold schedules for varying datasets.

### D.1. Gradient Concentration

Figure 13 plots the average side length of a square patch required to capture a fixed percentage of the gradient concentration heatmap for DDPM++, NCSN++ and DiT network denoisers. Although there are some differences in the gradient concentrations, the general trend is consistent that network denoisers exhibit a more locally concentrated gradient for small $t$, and less concentration as $t$ increases.

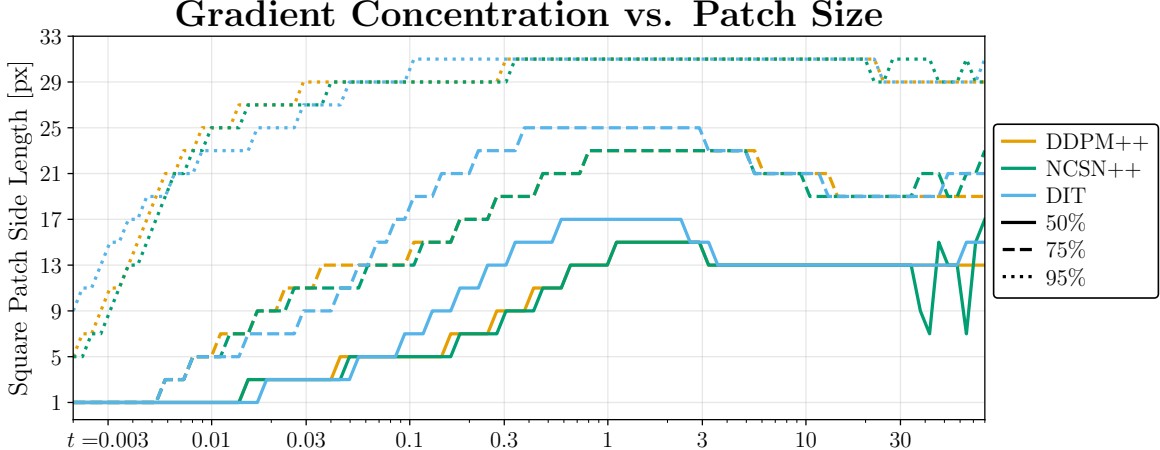

*Figure 13.* Comparison of the Gradient concentration of DDPM++, NCSN++ and DiT networks across varying diffusion time. All three network denoisers display the same anti-correlation between gradient concentration and diffusion time.

### D.2. Patch Errors

Figure 4 compares patch posterior means to patches of DDPM++ denoiser outputs for $\mathbf{z}$ drawn from that network's reverse process. In Figure 14, we produce similar plots for DDPM++, NCSN++, DiT and U-ViT network denoisers. Across each denoiser, patch posterior means have similar errors when compared to patches of the network output. At both small and large $t$, appropriately sized patch posterior means have low MSE against network denoiser patches, with poorer MSE over intermediate $t$. Additionally, for each network we observe a similar trend that as $t$ increases, network denoiser patches are best estimated by patch posterior means of increasing spatial size.

Notably, for U-ViT, we find that patch posterior means do not estimate high $t$ denoiser outputs well. This is a symptom of the amplification problem for $\epsilon$-predictor networks, described by (Karras et al., 2022).

### D.3. Denoiser Errors

Figure 7 visualizes the mean squared error between a variety of proposed denoisers and a DDPM++ network denoiser across the forward and reverse processes of three different datasets. However, other choices of network denoiser baseline are possible. Figure 15 plots denoiser MSE against four different network baselines. As checkpoints are only available on CIFAR-10 for DiT and U-ViT, no comparison is performed on FFHQ and AFHQ for these baselines. In addition to the denoisers shown in Figure 7, Figure 15 also plots MSE between pairs of network denoisers.

Examining Figure 15, PSPC-Flex consistently has lower MSE versus each network baseline than any other non-network denoiser. However, our method is consistently outperformed by the outputs of other network denoisers. We note that the gap between our approach and the performance of DiT when compared against DDPM++ and NCSN++ baselines is remarkably close. We believe closing this gap is a promising area of future research.

## E. Additional Denoiser Outputs

### E.1. CIFAR-10

We present additional examples of denoiser outputs for the same $t$ values presented in Figure 1. All $\mathbf{z}$ are drawn from the reverse process of the DDPM++ network denoiser.

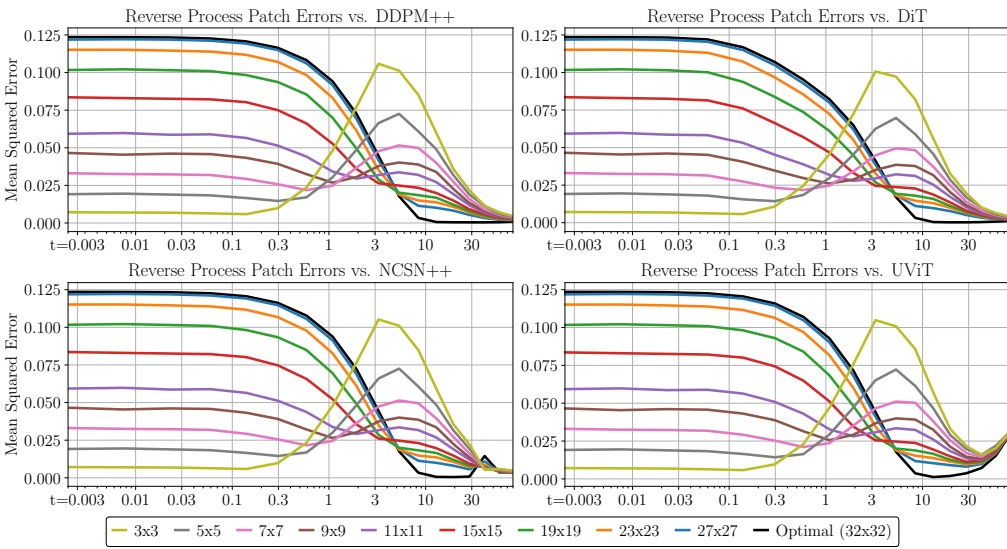

*Figure 14.* Comparison of patch posterior means of varying size against patches of network denoiser outputs for shared **z** drawn from each network's reverse process on CIFAR-10. Across each network, we observe that patch posterior means are better estimators of network output patches than the optimal denoiser.

## F. Additional PSPC-Flex Samples

We provide additional PSPC-Flex samples for CIFAR-10, FFHQ, and AFHQ datasets in Figure 24. For each row, the images in the left subplot has been generated from an identical latent seed from the corresponding image in the right subplot. In addition, Figure 25 compares samples from a larger set of denoisers, including PSPC-Flex, PSPC-Square, NCSN++, Gaussian, optimal, and CFDM.

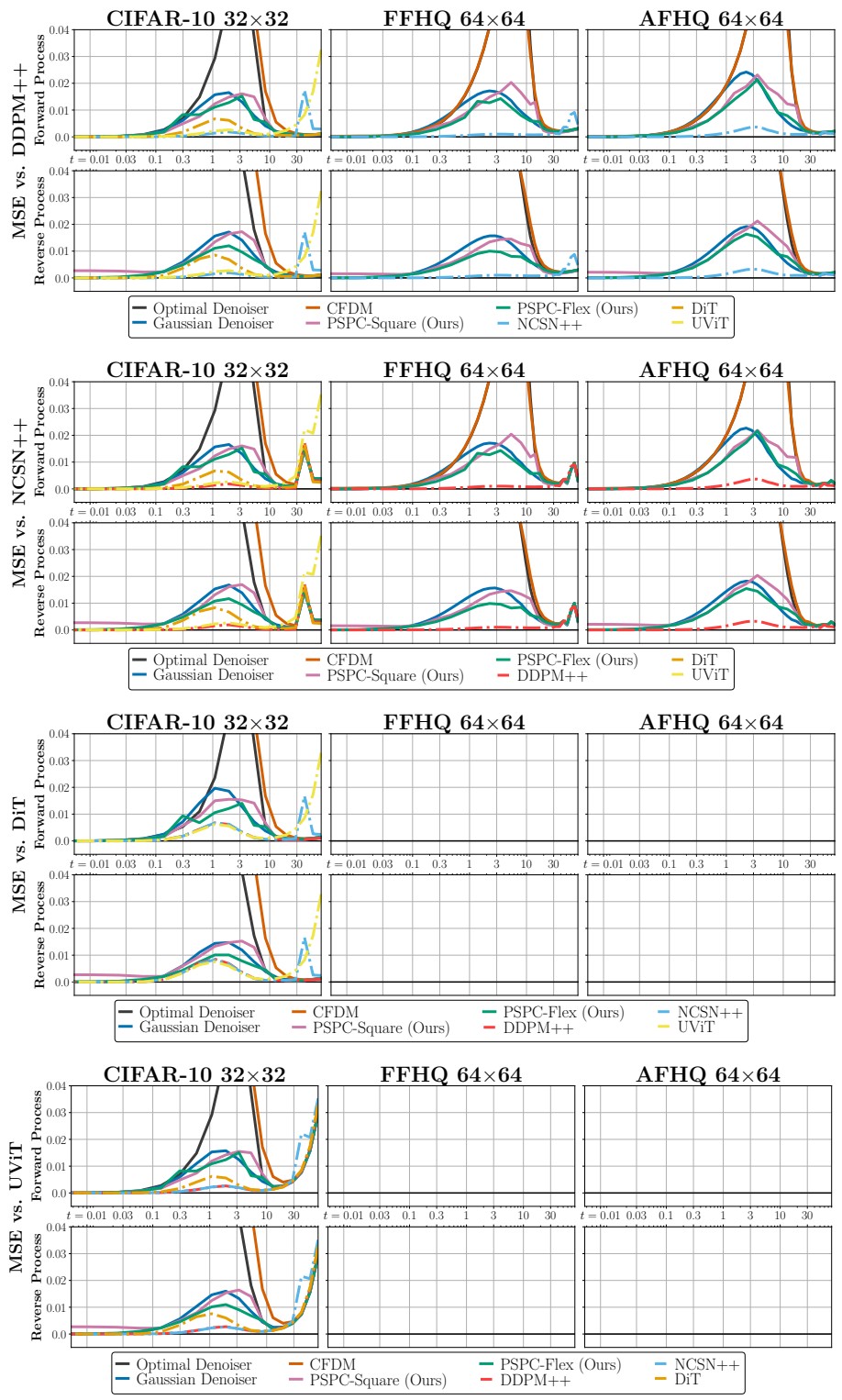

*Figure 15.* Comparison of network and empirical denoisers against varying network denoiser baselines. For DiT and UViT, no comparison is performed for FFHQ and AFHQ. Against each baseline, PSPC-Flex consistently outperforms Gaussian, CFDM, and optimal denoisers. However, PSPC performance consistently lags behind neural network performance.

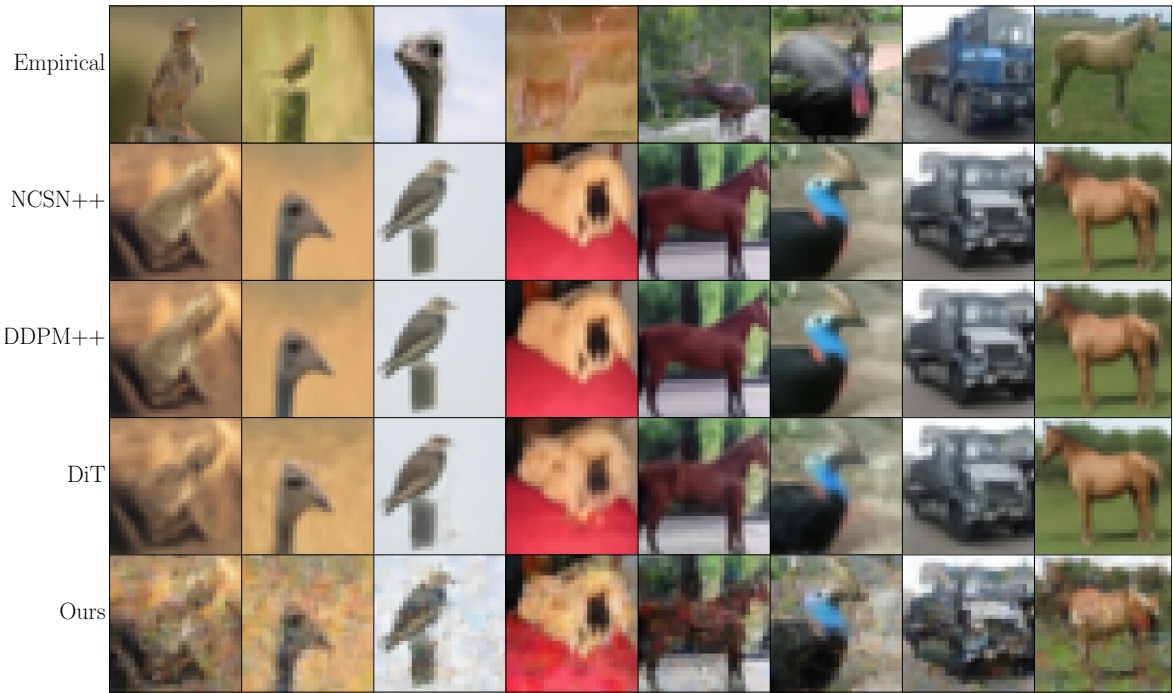

*Figure 16.* Additional CIFAR-10 denoiser outputs for $t = 0.3$

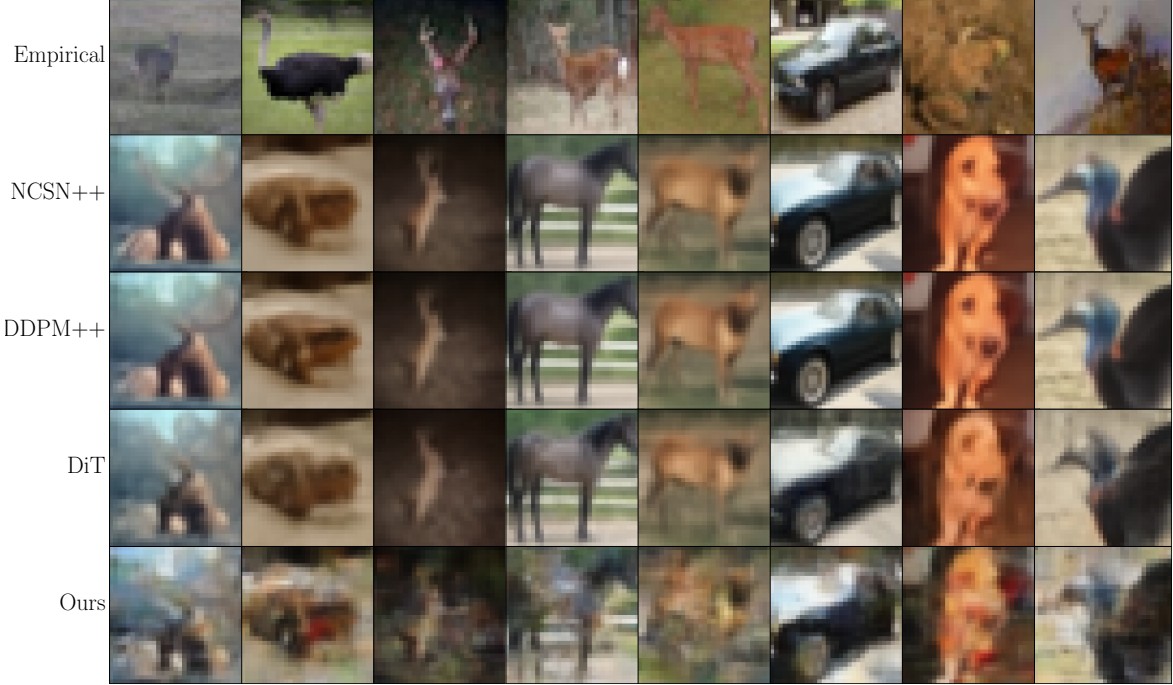

*Figure 17.* Additional CIFAR-10 denoiser outputs for $t = 0.6$

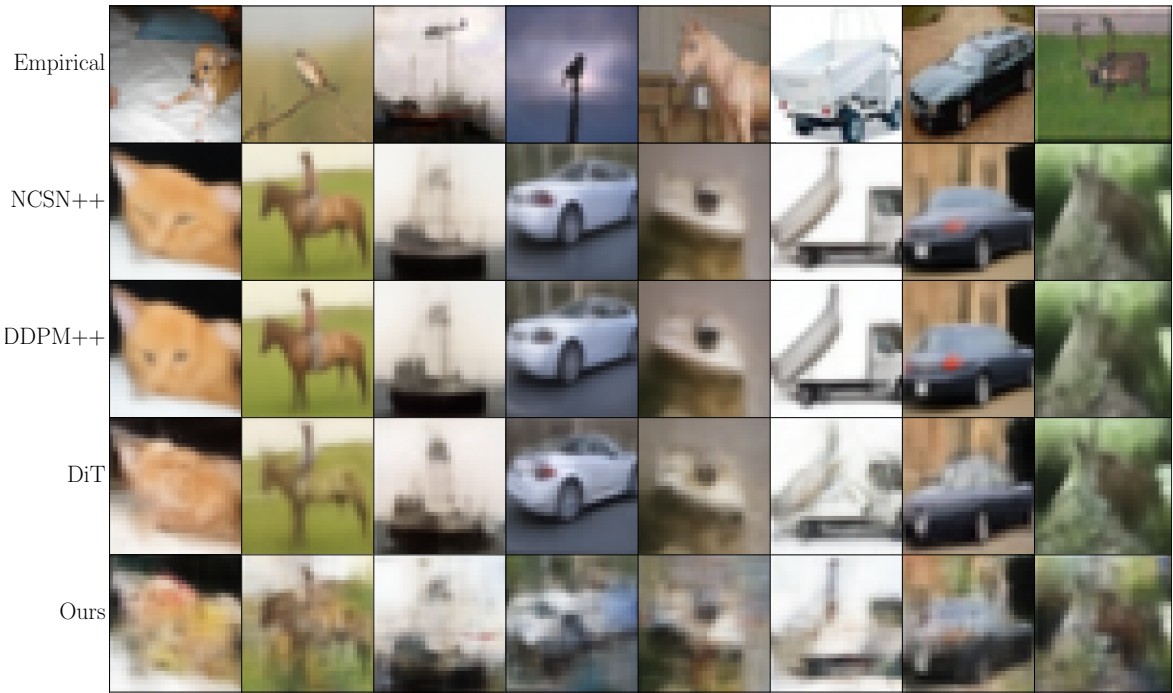

*Figure 18.* Additional CIFAR-10 denoiser outputs for $t = 1.1$

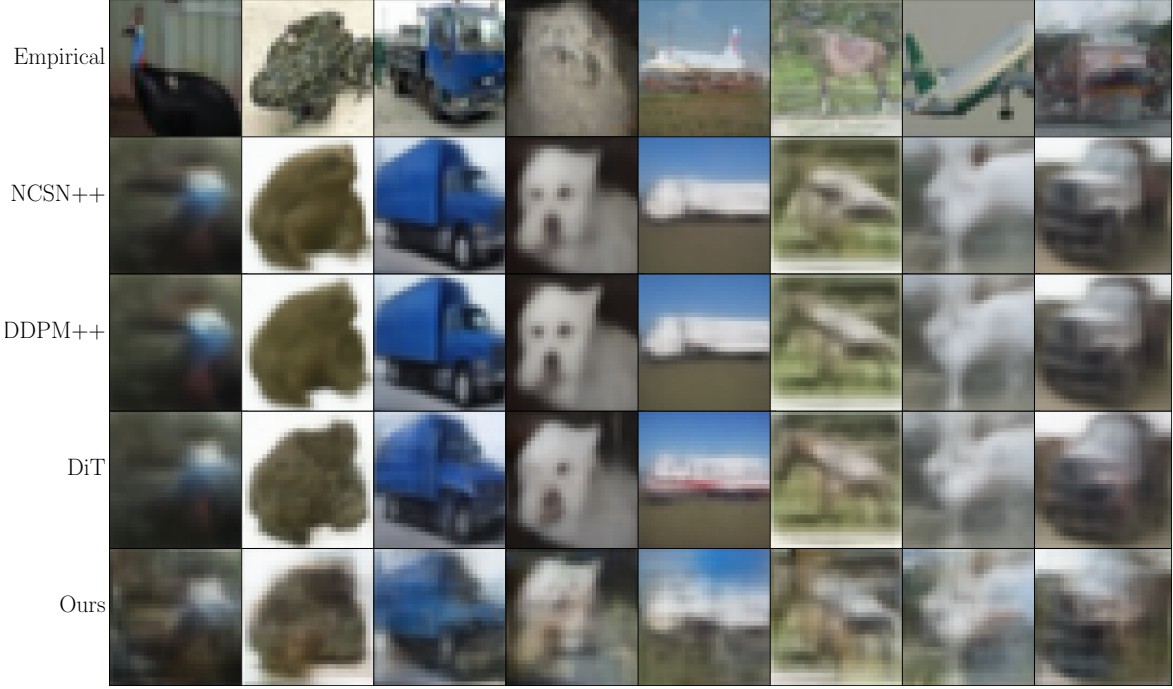

*Figure 19.* Additional CIFAR-10 denoiser outputs for $t = 1.9$

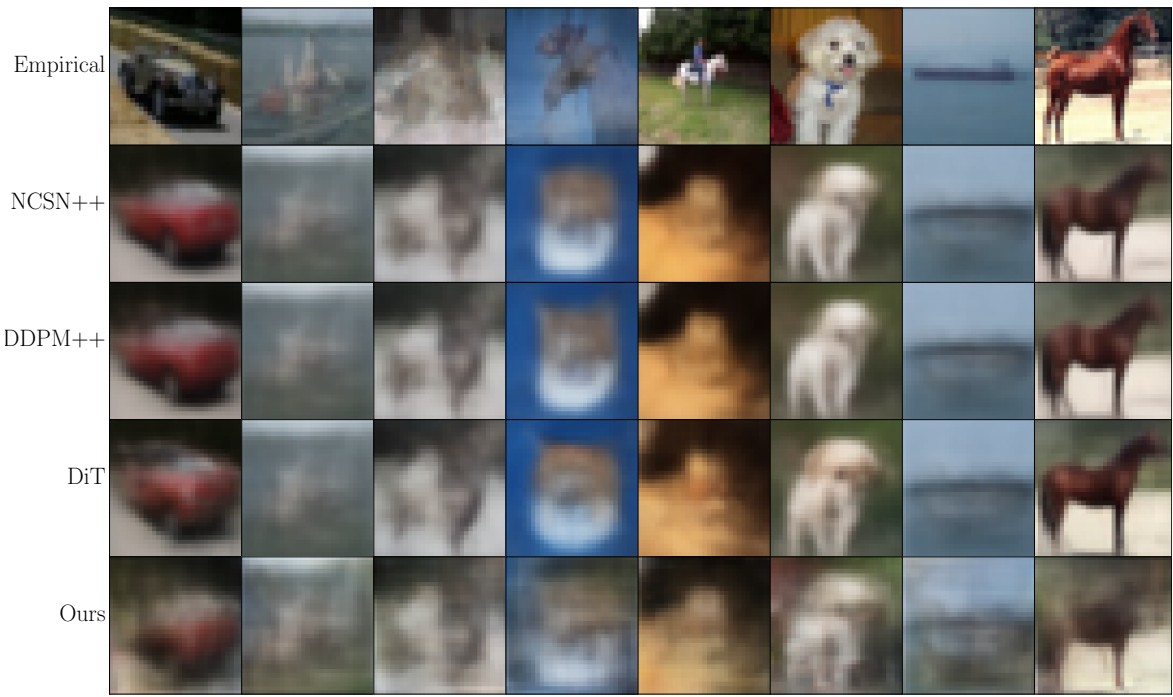

*Figure 20.* Additional CIFAR-10 denoiser outputs for $t = 3.3$

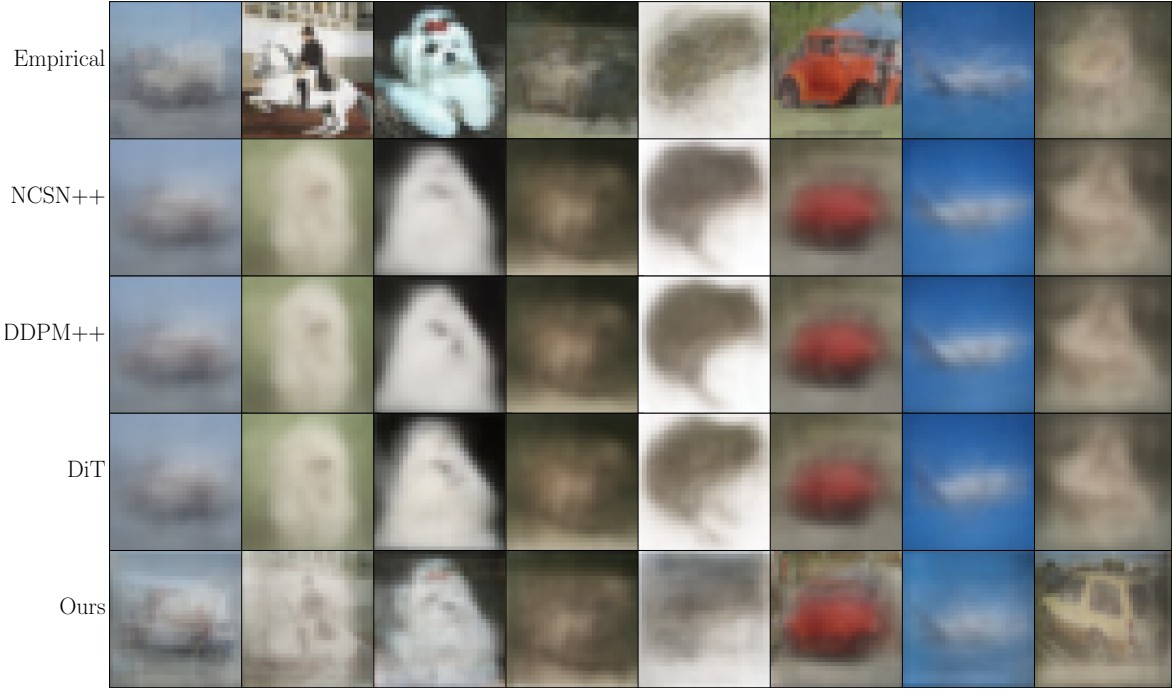

*Figure 21.* Additional CIFAR-10 denoiser outputs for $t = 5.3$

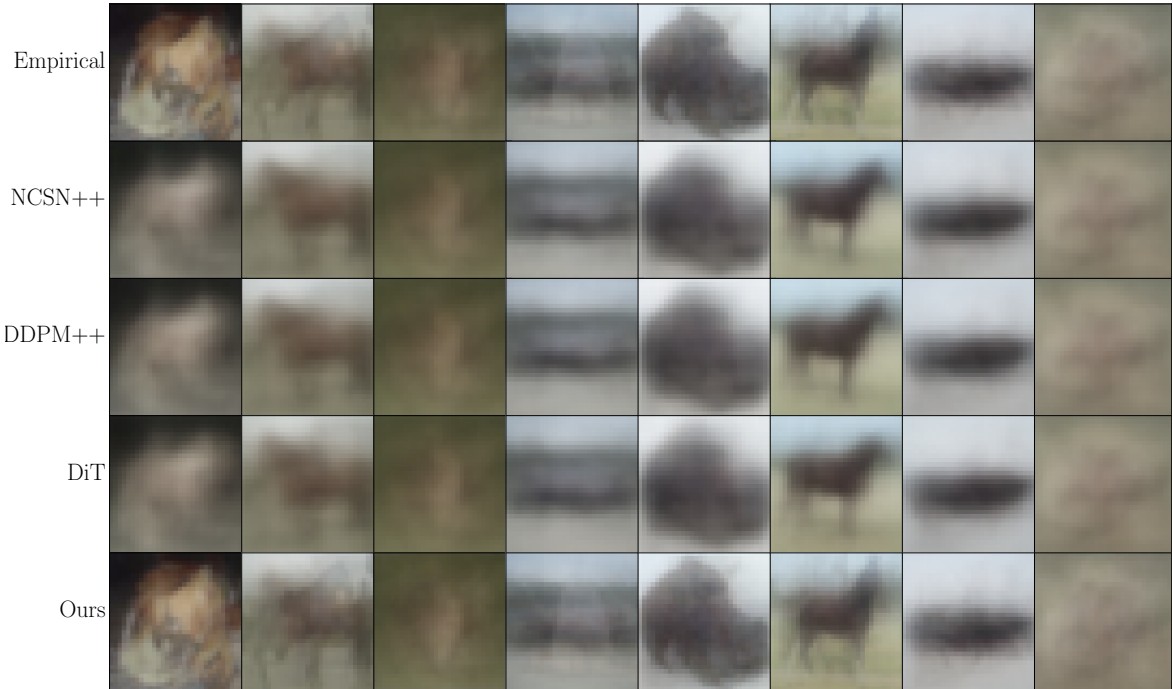

*Figure 22.* Additional CIFAR-10 denoiser outputs for $t = 8.4$

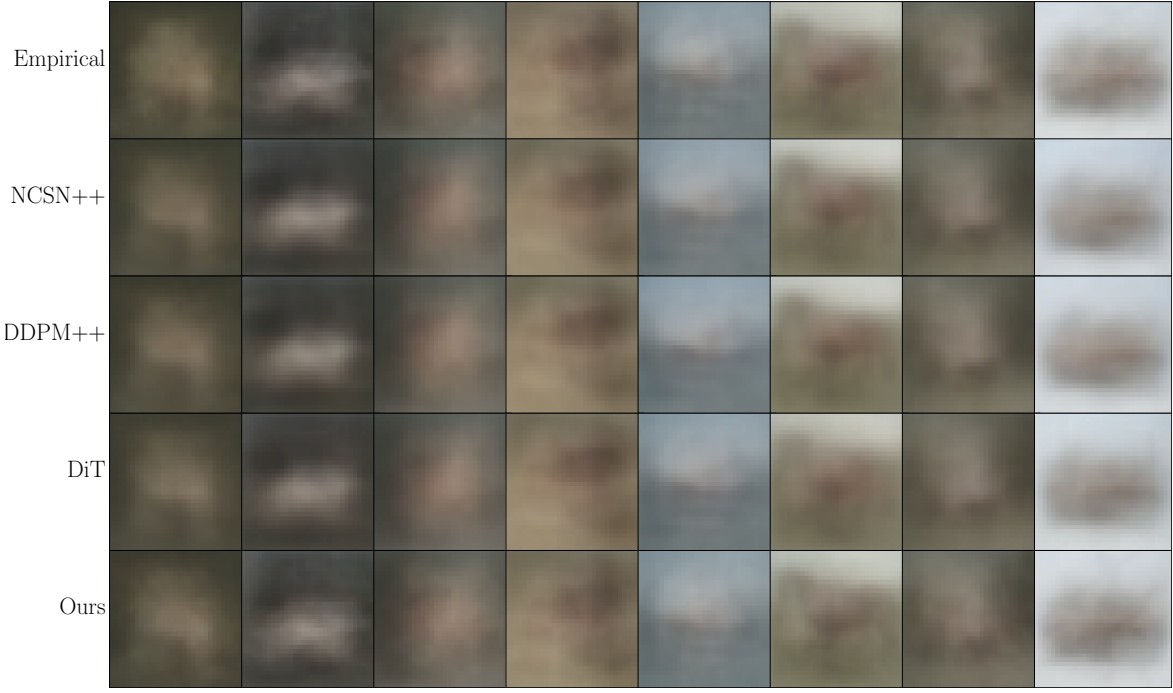

*Figure 23.* Additional CIFAR-10 denoiser outputs for $t = 12.9$

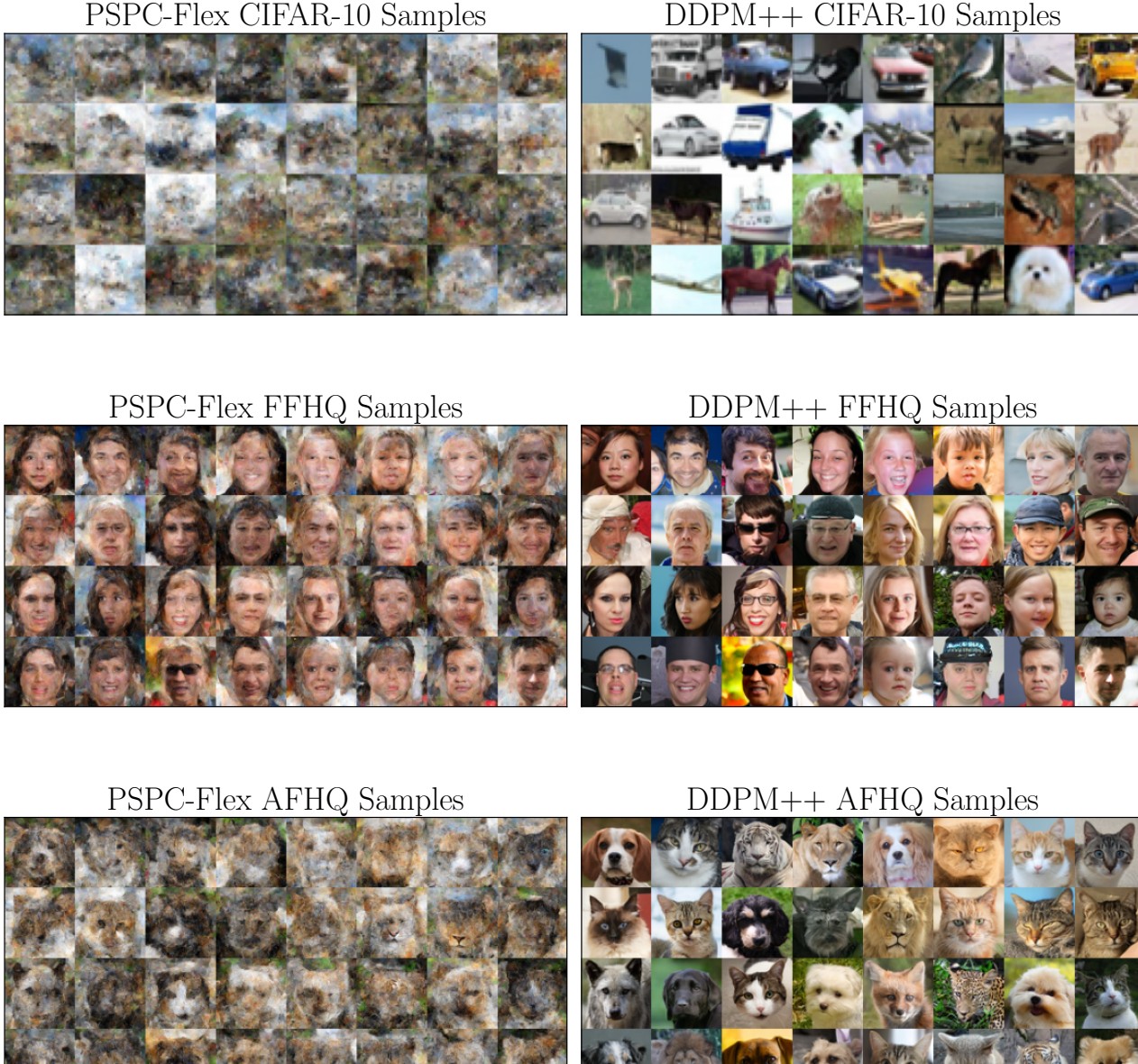

*Figure 24.* Additional PSPC-Flex PF-ODE samples on CIFAR-10, FFHQ and AFHQ. Samples in the left column are generated by PSPC-Flex, while those in the right column are generated by a DDPM++ network denoiser, starting from the same initial **z**.

## CIFAR-10 Samples - Shared Initial Conditions

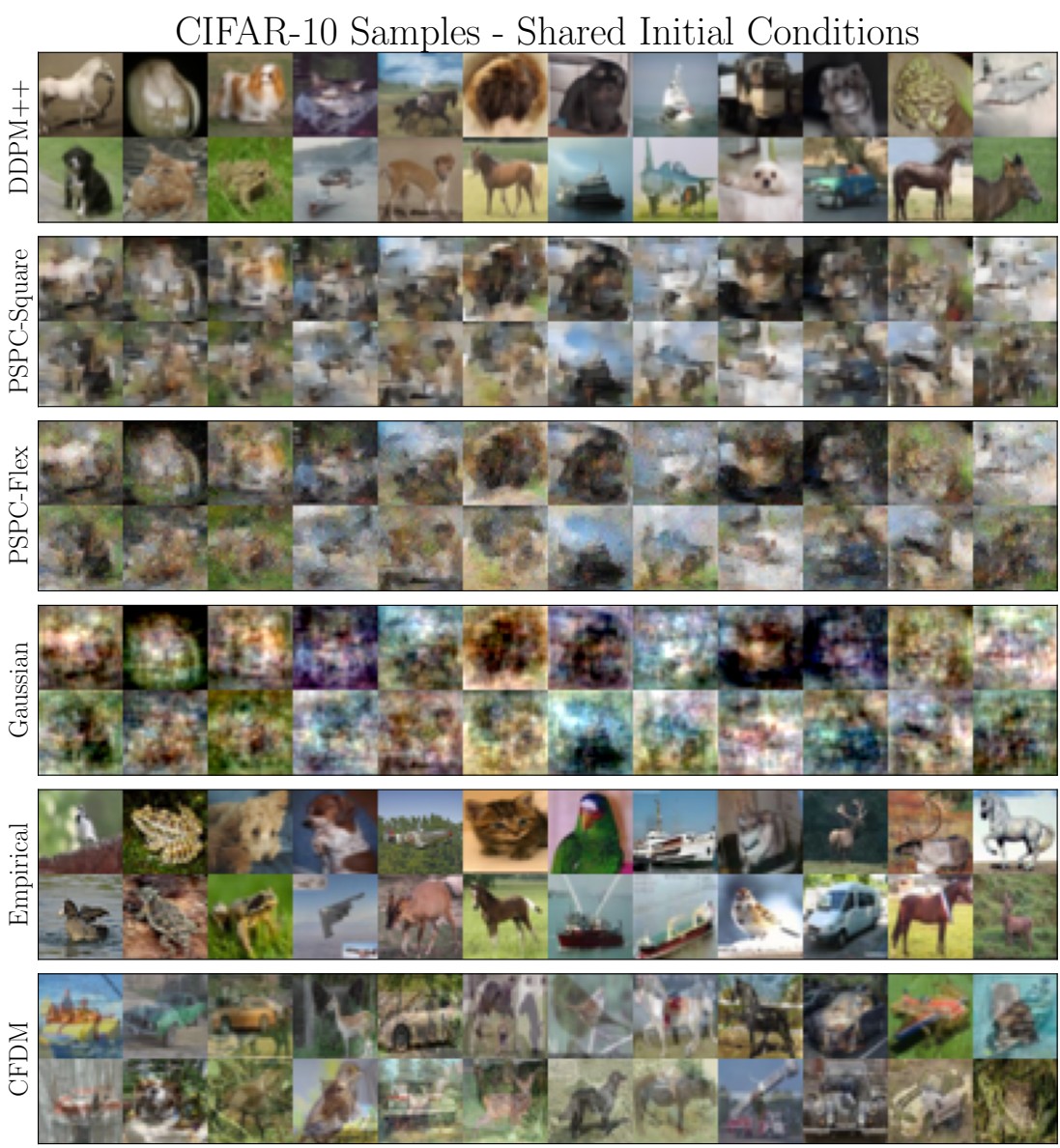

*Figure 25.* Comparison of PF-ODE CIFAR-10 samples generated using varying denoisers, starting from a shared initial **z**.

