# OpenReview forum: "Towards a Mechanistic Explanation of Diffusion Model Generalization"
_ICML.cc/2025/Conference — ICML 2025 spotlightposter_

### Official Review · Reviewer_6EJx · 2025-03-11

**Overall Recommendation:** 1

**Summary:**

In this paper, the authors analyze inductive biases of diffusion and relate these to the generalization capabilities of diffusion models. The authors start by examining the network denoiser approximation error, where the approximation error is defined as the deviation of the prediction from the optimal denoiser, where the optimal denoiser equals the weighted average over the images of the dataset. The authors analyze how this difference behaves for different architectures and find similar behavior across them. The authors proceed to show that the generalization of these models arises as a product of locally biased operations, showing this by approximating the operations using patch-based empirical denoisers. Finally, the authors propose PSPC, a training-free denoiser and show that PSPC and the other denoisers exhibit larger similarity between each other than that to the optimal denoiser, as well as that the samples produced with PSPC exhibit larger structural similarities.

**Claims And Evidence:**

I have several significant concerns regarding the paper's claims and methodology. Firstly, I found that many of the authors' assertions are unsupported by evidence or lack clear definitions. One of the primary motivators of the work is the concept of "local bias," but the authors fail to define or reference this term. This omission makes it challenging to understand the paper's main arguments and results.






In the following section, the authors do not really provide sufficient motivation and the "local bias" is not properly introduced or referenced. Furthermore, as the main claim of the paper is that this inductive bias is similar across various architectures and noise schedulers, it is disappointing that the authors have performed the remaining experiments only using DDPM++. I did not find any analyses corresponding to other architectures in the supplementary material either.

The biggest concern I have is that many of the effects which the authors analyze (e.g., gradient heatmaps and local behaviour) might be coming from the fact that all four chosen architectures use attention mechanisms. In a related work (which the authors correctly point out) by Kadkhodaie et al. (2023), it is suggested that generalization occurs as the inductive biases of networks interplay well with geometrically adaptive harmonic bases.

**Essential References Not Discussed:**

I did not find any references to be missing, although I believe that the work by Kadkhodaie et al. (2023) should've been referenced much earlier as they provide an important and significant contribution to explanation of the generalization phenomena in diffusion models.

**Experimental Designs Or Analyses:**

Please see above.

**Methods And Evaluation Criteria:**

I believe that the problem of understanding generalization in diffusion models requires either more rigorous approaches (from a theoretical perspective) that can be backed up by numerical experiments, or in the case of pure numerical experiments (such as in this paper), much larger and stronger set of evaluations.

For example, the methodological issues in Figure 2 raise concerns: the authors display the MSE vs . Optimal Denoiser only for time up to t=30, but it is visible that for both U-ViT and NCSN++ the differences start to increase after t=30. Additionally, the averaging of 10,000 samples might suppress differences between the networks. To address this, the authors could focus on averaging paths that are semantically similar, such as averaging only 100 paths that generate an image from a same class, where the class has been obtained using a pre-trained classifier. This would help reduce the "averaging effect" and provide a more accurate representation of the similarities of the models' behavior.

Furthermore, the following section lacks sufficient motivation for the "local bias" concept, and the authors do not provide adequate analysis or references to support their claims. Moreover, the main claim of the paper – that the inductive bias is similar across various architectures and noise schedulers – is not adequately supported by experiments, as all experiments past Figure 2. only use DDPM++.

Also, the effects analyzed in the paper (e.g., gradient heatmaps and local behavior) might be attributed to the fact that all four chosen architectures use attention mechanisms. A related work by Kadkhodaie et al. (2023), which the authors correctly cite, suggests that generalization occurs due to the interplay between inductive biases and geometrically adaptive harmonic bases. In my opinion, this paper provides a stronger and cleaner argument towards the inductive biases. But, more importantly, the authors performs analysis using a UNet architecture, as well as BF-CNN, a version of DNCNN network, neither of which use attention. In my opinion, it would be crucial to include either of these architectures (or both) , in order to be able to argue that the methodology in sections 4 and 5 can actually be attributed to generalization and not just attention.

**Other Comments Or Suggestions:**

1. It is unclear why the authors included the sentence "Corroborating findings of Niedoba et al…": what are the findings that the authors are referring to here and why is this paper relevant to the problem at hand? The authors should point this out when citing it.
2. As mentioned previously, in Section 3.1, the authors analyze "local bias" without defining it or giving a reference to the definition. I am not aware of what do they mean by this, and this really dampens the strength of the motivation for the section, as well as the subsequent ones.
3. I think that the authors should include what does "near SOTA" mean, and provide some performance of their trained models (e.g., please provide FID) to strengthen the validity of their experiments.
4. What do the authors mean in line 152 when they write “of the network denoiser output at pixel (x,y)”? What is pixel "(x,y)"?
5. a) In equation (8), shouldn’t the first argument of the network D_theta be z and not x?
5. b) In same equation, what is the notation x,y,c (in the subscript) supposed to represent? Also, why is there y in equation (8), as from their introduction of the network they use is unconditional and does not take into account information y?
6. Finally, what do the authors mean by "drawing 10000 z samples from the forward process" (around line 150)? Do they mean sampling 10000 z samples and running forward process of 150 steps each time?

**Other Strengths And Weaknesses:**

Please see above.

**Questions For Authors:**

Please see above.

**Relation To Broader Scientific Literature:**

Although the phenomena of diffusion's ability to generalize is of great interest, I find that the contribution of this paper lacks, as well as the validity of their claims.

**Theoretical Claims:**

There were no proofs or theoretical results in this paper.

---

> ### Author Rebuttal · Authors · 2025-04-01
>
> ## General Comment
> We'd like to thank the reviewer for their thorough review. We have summarized and responded to the key points of your review below in the limited available space. If we have missed any of your points, we welcome further discussion.
> ## Methods & Evaluation
> **Network errors for  $t>30$**
>
> The reviewer is right. U-ViT & NCSN++ MSE is higher for $t> 30$. As U-VIT predictis $\epsilon$, we compute $x$ as $x = z - t \epsilon_\theta(z, t)$. This scales $\epsilon_\theta$ errors by $t$ (see L162). NCSN++ errors fluctuate near $t=40$, but are ~4x smaller than peak errors at t=3. We believe this is a training artifact. Further, Fig. 9 implies high-$t$ errors have little impact on generalization. Both U-ViT& NCSN++ have SSCD similarities > 0.7 with every model, indicating near-copy samples.
>
> **"Averaging effect"**
>
> Beyond MSE, Fig. 1, 2, & 13-20 provide evidence that network outputs are qualitatively similar for the same input. To quantify this output similarity, we have measured output cosine similarity at $t=3.2$ for shared forward process inputs [[Link](https://drive.google.com/file/d/1hk6b-m7meFDw6qjw70_579REIpevgLIr)].
>
> **Lack of other networks baselines past Fig. 2**
>
> This assertion is incorrect. Past Fig 2, Fig 9 and 13-20 all compare various models. However, for transparency and completeness, we will include the following modified versions of Fig 3, 4, and 6 to the appendix:
> - [Fig 3](https://drive.google.com/file/d/1U1x3id-8ECHJugKNPVMouBRyHho4tpsB/) All models have a similar trend between gradient concentration and $t$.
> - [Fig. 4](https://drive.google.com/file/d/1qLbefgMe_FI_dGph9LRPLSbZLcXGamHn/) Patch posterior MSE is similar for all models, except high-$t$ U-ViT.
> - [Fig. 6](https://drive.google.com/drive/folders/1_296T4B96x_r7xAzczftO5YB9PlLZ0F8). We compare our methods against each network baseline. For UViT & DiT, we only have CIFAR checkpoints.
>
> We are happy to provide additional figures upon request.
>
> **On the impact of attention**
>
> We do not believe attention layers are the cause of our observations. To test this, we trained an attention-free DDPM++ model on CIFAR. We find the MSE of PSPC is mostly unaffected[[Link](https://drive.google.com/file/d/1H6UrOp4looPt9m8FAVtcuvn4YlGW0Tu9)]. Qualitatively, the attention-free model's outputs are similar to other models [[Link](https://drive.google.com/file/d/1fnjmlGHC77uiXUYaJkOJI2cMkkDmSyO1)]. Similar to our work, Kamb & Ganguli find their patch-based denoiser performs **better** against attention free models.
>
> **On the inclusion of U-Net and BFCNN**
>
> This paper's aim was to investigate generalization in modern image diffusion models, which overwhelmingly utilize attention layers. In addition to the lack of attention, the networks of Kadkhodaie et al. (2023) have unique properties such as no bias terms, and no $t$ conditioning. Due to these architectural quirks, we believe that analyzing commonly used U-Net and DiT architectures is of greater value to the community than BFCNN.
>
> ## Other Comments
> **Q1**
>
> Niedoba et al. (Fig. 3) finds network denoisers are biased for $t \in [0.3, 10]$, which matches our Fig 2 findings. For clarity, we suggest the change:
> >Across all architectures, we observe similar behaviour to Niedoba et al. (2024), Figure 3 - network denoisers exhibit low MSE for both small and large values of $t$, but substantial error for $t \in [0.3, 10]$.
>
> **Q2**
>
> We use "local inductive bias" to describe a preference for denoising functions where outputs at a spatial position (x,y) are more influenced by nearby input pixels than distant ones. Goyal & Bengio (2022) define inductive bias as a tendency for learning algorithms to favour solutions with certain properties. Kamb & Ganguli (2024) provide a more formal definition of the bias, calling it "locality." However, since Sec 3.1 mainly motivates our hypothesis on generalization via local denoising, we felt a formal definition was unnecessary. We propose the following changes:
> - Replacing “local bias” with “local inductive bias” throughout
> - Rephrasing the intro of Sec. 3.1 (L142) as: “One potential inductive bias (Goyal & Bengio 2022) of network denoisers is local inductive bias, where denoiser outputs are more sensitive to spatially local perturbations of $z$ than distant ones”
>
> **Q3**
>
> The networks analyzed have high performance but are not SoTA. Here are their FID scores
> |Model|CIFAR|FFHQ|AFHQ|
> |-|-|-|-|
> |DDPM|1.97| 2.39|1.96|
> |NCSN|1.98| 2.53|2.16|
> |DiT|9.08|||
> |UViT|3.11|||
>
> **Q4**
>
> Pixel (x,y) is the output pixel at spatial location x,y where $x \in \\{1, ..., w\\}, y \in \\{1, ..., h\\}$ for image height $h$ & width $w$. We'll update this notation for clarity
>
> **Q5**
>
> We'll fix this error.
>
> **Q6**
>
> The subscript x,y,c denotes indexing at position (x,y) and channel $c \in \{1, 2, 3\}$. We'll reword this to improve clarity.
>
> **Q7**
>
> For each $t$ value, we draw 10k samples from the forward process $z \sim p_t(z | x^{(i)}) p_D(x)$, with $p_t(z | x^{(i)})$ defined on L75.

---

> > ### Comment · Reviewer_6EJx · 2025-04-02
> >
> > I would like to express my gratitude to the authors for their comprehensive response, particularly in light of the additional network that does not incorporate attention. I appreciate the effort they have put into addressing the concerns raised.
> >
> > However, I still have many reservations about the work. I agree with reviewer wsKg that the approach and insight into analyzing diffusion through locality bias are indeed intriguing. Nevertheless, several questions remain unanswered:
> >
> > - Is the CIFAR dataset sufficient to support these claims, or is it too simplistic (specifically in terms of non-attention networks)?
> > - Is the observed locality a consequence of the dataset's simplicity, or can this claim be generalized to more complex datasets with intricate local and global structures?
> > - If the generalization phenomenon is indeed linked to local denoising, why is it that the proposed method performs so poorly even on the simple datasets considered?
> >
> > To strengthen the paper's claims about generalization, I firmly believe that more extensive and rigorous analysis is necessary, especially if the authors rely solely on empirical evidence. These concerns still make me believe that that the paper falls short of acceptance at this time.
> >
> > I would like to acknowledge that my expertise in the literature may not be as comprehensive as that of other reviewers, and I kindly request that the Area Chairs take this into consideration when evaluating my feedback.
> >
> > However, as a researcher who has been actively working on advancing our understanding of the generalization capabilities of diffusion models, I must confess that I remain unconvinced by the paper's arguments. My reservations stem from my own experiences and insights gained through working closely with these models, and I hope that my feedback will be taken in the spirit of constructive criticism.

---

> > > ### Author Response · Authors · 2025-04-05
> > >
> > > ### **General Comment**
> > > Thank you to the reviewer for engaging with our rebuttal. We appreciate the opportunity to further discuss why we believe our work warrants acceptance. In our previous response, we made a concerted effort to individually address each component of the initial review. Of the issues we initially addressed, only the impact of self-attention was mentioned in the rebuttal reply. We hope this indicates that our responses in the other areas satisfactorily resolved the reviewer’s concerns.
> > >
> > > However, in light of the reviewer’s unchanged evaluation, it remains unclear whether our responses did not sufficiently address the original concerns, or whether the addressed concerns were not central to their overall assessment. In either case, more specific feedback would have been appreciated to better understand the basis for the continued reservations, so that we might have addressed them more effectively.
> > >
> > > The rebuttal reply also introduces three new concerns not mentioned in the original review. While we are, of course, happy to address them here, we would have welcomed the opportunity to engage with them earlier in the review process.
> > >
> > > ### **Response to Additional Questions**
> > >
> > > > - Is the CIFAR dataset sufficient to support these claims, or is it too simplistic (specifically in terms of non-attention networks)?
> > >
> > > We would like to clarify that along with CIFAR, our work also evaluates PSPC on  FFHQ and AFHQ which are standard in the field. In the literature, these datasets have been used in similar prior art to support arguments about generalization in diffusion models. For example, the analysis of [1] predominantly relies on CIFAR, those of [2] mostly use CIFAR & FFHQ, and [3] utilizes the same datasets as our work.
> > >
> > > We believe Figure 6 (including the alternative versions included in our rebuttal) clearly illustrates that PSPC methods have quantitatively similar relative performance to other denoisers on each dataset. Figure 8 further highlights the qualitative similarities between PSPC and Network denoisers across these datasets.
> > >
> > > In terms of non-attention networks, the performance of DDPM++ and the attention-free alternative we trained on CIFAR are nearly identical. We have no reason to believe that this relationship would be substantially different for FFHQ or AFHQ. Unfortunately, we are unable to retrain a attention free DDPM++ variant on these datasets in the time remaining in the discussion period.
> > >
> > > [1] Zhang. J. et al, “The emergence of reproducibility and consistency in diffusion models” 2023
> > >
> > > [2] Wang B. and Vastola, J.J. “The unreasonable effectiveness of gaussian score approximation” 2024
> > >
> > > [3] Li X. et al. “Understanding generalizability of diffusion models requires rethinking the hidden gaussian structure.” 2024
> > >
> > > > - Is the observed locality a consequence of the dataset's simplicity, or can this claim be generalized to more complex datasets with intricate local and global structures?
> > >
> > > In addition to our comments previously regarding the widespread usage of our chosen datasets, we would also like to mention that all three datasets have both local and global structure. In both AFHQ and FFHQ, there are global structures such as facial shape and pose, as well as local structures such as hair texture. In CIFAR, this is somewhat diminished by the resolution. However, Fig. 2 still clearly shows a mix of global and local structure - a kitten, with globally positioned ears, tail and paws, with local coat colourations.
> > >
> > > > - If the generalization phenomenon is indeed linked to local denoising, why is it that the proposed method performs so poorly even on the simple datasets considered?
> > >
> > > We respond to this point in the first part of our response to Q1. of reviewer **wsKg**. To elaborate, while it is true that PSPC’s performance is poor compared to network denoisers, it is - to the best of our knowledge - the best available empirical approximation to the output of network denoisers. We believe that this similarity is evidence to reasonably conclude that local denoising is one piece of the diffusion generalization puzzle. However, it is apparent from the differences that there are still missing pieces remaining. We believe that understanding what these remaining missing pieces are is an important and exciting research direction for the community.

---

### Official Review · Reviewer_wsKg · 2025-03-11

**Overall Recommendation:** 3

**Summary:**

The paper studies the inductive bias of diffusion models that enable generalization. The authors attribute such inductive bias to the locally denoising capability of diffusion models, which is supported by the observation that the network outputs are sensitive to the local perturbation of its input.

Based on this intuition, the authors then propose to reproduce the generalization of diffusion models with a patch-based local denoiser. They find that the resulting denoiser generates similar outputs to those of the real diffusion models.

**Claims And Evidence:**

The paper claims that the PSPC-Flex samples are remarkably similar to those of the diffusion model. However, Figure 8, 21 and 22 shows the generated images of PSPC-Flex is significantly different from those generated from the diffusion model, especially for Figure 21. This implies the locally-denoising inductive bias cannot well explain the inductive bias of real diffusion model.

**Essential References Not Discussed:**

I am not aware of any essential references that are not discussed.

**Experimental Designs Or Analyses:**

I have doubt on measuring the generalization ability of different models using the MSE of denoiser outputs. The detailed reason can be found in the "Methods And Evaluation Criteria" section.

**Methods And Evaluation Criteria:**

The authors demonstrate the similarity between the proposed patch denoiser and the real diffusion models by comparing the denoiser outputs, which might not be equivalent to the generalization capability. For example, although Figures 13-20 demonstrate PSPC can generate similar denoiser outputs as those of the diffusion models, PSPC fail to generate high quality samples (Figure 21). This suggests there exists a huge gap between the generalization ability of the models and the denoising ability of the models. Due to this gap, the authors should propose alternative metrics for measuring the generalization ability.

Another issue is the authors only compare PSPC with the Gaussian denoiser. More baselines, such as the closed-form diffusion models and other types of patch-based diffusion models in the paper should be included.

**Other Comments Or Suggestions:**

Are the labels in Figure 8 correct? Which row corresponds to DDPM++ ?

**Other Strengths And Weaknesses:**

No comments.

**Questions For Authors:**

1. Please clarify the gap between denoising ability and generalization ability. Why the proposed patch denoiser well-matches the diffusion models in Figure 13-20 but diverges significantly in Figure 21? Such gap is not adequately addressed in the current paper. This raises concerns on how much the locally denoising mechanism contributes to the generalization ability.

2. Please compare the proposed denoiser with more (classical patch based) denoisers.

3. Why PCSC shares the highest similarity with the Gaussian denoiser rather than the diffusion model (Figure 9)?

**Relation To Broader Scientific Literature:**

Previous works have shown that diffusion models have certain inductive bias that enable generalization. The authors try to characterize the properties of such inductive bias. They attribute such inductive bias to the locally denoising operation. The findings are novel.

**Theoretical Claims:**

The theoretical claims have no obvious issues.

---

> ### Author Rebuttal · Authors · 2025-03-31
>
> ### **General Response**
>
> We'd like to thank the reviewer for taking the time to read and review or work. Please find our responses to your review below. If there are any items which you believe have not been addressed, we welcome this feedback.
>
> ### **Other Comments Or Suggestions:**
> > Are the labels in Figure 8 correct? Which row corresponds to DDPM++ ?
>
> The labels in Figure 8 are correct, the middle row of each subplot corresponds to DDPM++ (An EDM architecture) evaluated on the $\mathbf{z}$ value in the top row. The left column of subplots show sampling trajectories using the DDPM++ denoiser, while the right column shows sampling trajectories using the PSPC-Flex denoiser.
>
> ### **Questions For Authors:**
>
> > 1. Please clarify the gap between denoising ability and generalization ability. Why the proposed patch denoiser well-matches the diffusion models in Figure 13-20 but diverges significantly in Figure 21? Such gap is not adequately addressed in the current paper. This raises concerns on how much the locally denoising mechanism contributes to the generalization ability.
>
> The reviewer is correct that the samples produced by PSPC are significantly different to those produced by network denoisers. Although we tried our best to match the output of network denoisers as closely as possible, our method is an imperfect approximation to their behaviour. There is a significant difference between our fully empirical method and a deep neural network trained with gradient descent over millions of $\mathbf{z}$ samples. We believe the remarkable similarity of our method our denoiser presents compelling evidence that network denoisers may employ similar generalization mechanisms.
>
> Figure 1 & 6 illustrate that significant differences remain between PSPC and network denoisers, especially for intermediate $t$. When sampling with PSPC-Flex, these differences in denoiser outputs compound to result in PSPC-Flex PF-ODE trajectories which slowly drift from the PF-ODE trajectories of network denoisers. We discuss this briefly in section 6.2, line 371.
>
> This process of error accumulation is visualized in Figure 8 which illustrates the denoiser outputs of PSPC-Flex and DDPM++ on shared $\mathbf{z}$ inputs drawn from  DDPM++ (left) and PSPC-Flex (Right) PF-ODE trajectories. Comparing the network and PSPC denoisers, we can see that for all cases and $t$, the outputs are highly similar. However, differences in two, which are most pronounced in the middle of the trajectory, result in substantially worse samples in the right column of figure 8 (PSPC-Flex) than the left column (DDPM++). Despite the artifacts present in PSPC samples, we highlight that the structure of the final samples in the right column of Figure 8 are similar to the final network samples in the right column.
>
> Notably, even as sample quality degrades (ie right column, t < 0.5) both denoisers produce similar denoiser outputs. These degraded $\mathbf{z}$ samples are obviously outside the training distribution $p_t(\mathbf{z})$ and are clear examples of neural network generalization. The fact that PSPC-Flex outputs are similar to network denoisers in this case provides further evidence that a local denoising mechanism may be partially responsible for the generalization of diffusion denoisers.
>
> > 2. Please compare the proposed denoiser with more (classical patch based) denoisers.
>
> In the methods & evaluation criteria section of your review, you specifically mention Closed-Form Diffusion Models (Scarvellis et al. 2023) as a missing baseline method. We have therefore implemented this method and added it as a baseline to Figure 6 [here](https://drive.google.com/file/d/1esmDa7DP2rPNjPShzRCh8UTPUUMzauPe/view?usp=sharing). We used hyperparameters $\sigma=0.1, M=2$, which they report in Appendix section C2. for their CelebA results. However, we did not find that CFDM outputs were significantly different to the optimal denoiser with these settings.
>
> If there are other baselines you believe are necessary, please cite them explicitly and we will try our best to implement them by the end of the discussion period.
>
> > 3. Why PCSC shares the highest similarity with the Gaussian denoiser rather than the diffusion model (Figure 9)?
>
> We are unsure as to why PSPC-Flex samples are more similar to the Gaussian denoiser. Recently, several methods have been proposed to explain diffusion generalization, including our work, the optimal Gaussian denoiser, and the geometrically adaptive harmonic bases of Kadkhodaie et. al (2023). Understanding the similarities between these methodologies is an interesting direction for future research.

---

> > ### Comment · Reviewer_wsKg · 2025-04-02
> >
> > I appreciate the authors response. Below,  I will add my additional comments.
> >
> > 1. My concern on the choice of using denoising outputs to measure the similarity between diffusion denosier and the proposed denoiser since though the proposed method produce highly similar denoising outputs are similar, the generated quality is much worse. Since we are interested in using the model to generate images, not perform denoising, I think the evaluation in the paper should focus more on the generating ability. I would like to see the authors evaluating the proposed method using metrics like FID, IS, Recall, Precision, which are commonly used for evaluating generative models.
> >
> >  2. Related to my first point, since the authors evaluate the similarity between proposed denoisers and diffusion models using the denoising outputs, I am interested in whether some traditional patch-based denoisers can generate images as well. For example, can BM3d generate images that are very close those of diffusion models?
> >
> > 3. It is still kind of strange to me that PCSC is closer to the Gaussian denoiser compared to the actual diffusion models since it seems the Gaussian denoiser does not use the locality inductive bias. It would be nice if the authors can provide some visualizations of its samples and compare with those of PCSC. Similarly, I'd like to see the generated samples of traditional method such as BM3d and the references mentioned in section 7, patch-based denoising section (it is sufficient to just show me one or two algorithms). I would also appreciate it if the authors can visualize the samples generated by the closed-form diffusion algorithm.
> >
> > #######################################################################################################
> >
> > Overall I think this work brings novel and interesting insights (locality bias) into the current progress of understanding diffusion generation. But the way the locality is implemented by PCSC might not be optimal as the generated samples do not look good enough on Cifar-10. Though I believe locality is an important inductive bias of diffusion model, it remains unclear how much locality contributes to the generalization. That being said, I do think this work is meaningful and worth being published.

---

> > > ### Author Response · Authors · 2025-04-07
> > >
> > > We'd like to thank the reviewer for their consideration of our rebuttal. We also appreciate their praise that our work is meaningful, novel, interesting, and worthy of publication. Below we have addressed each item of the reviewer's additional comments.
> > >
> > > ### **1. Sample Quality Metrics**
> > > We slightly disagree with the reviewer’s statement:
> > > > Since we are interested in using the model to generate images, not perform denoising
> > >
> > > While this is certainly true when designing a generative model, this was _not_ our goal when designing PSPC. Instead, our aim was to understand how diffusion models generalize. As diffusion samples are the result of sequential denoiser generalizations, we believe that individually analyzing each step of the sampling procedure is needed to understand the generalization process as a whole. In this context, we believe PSPC’s “highly similar denoising outputs” provide strong evidence to support our hypothesis that local denoising is a key component of this generalization process.
> > >
> > > We fully agree with the reviewer that the implementation of locality in PSPC is almost certainly not optimal. As it stands, the sample quality of PSPC is too poor to be used as a generative model. In the future, we believe understanding the autoregressive drift mentioned in our original rebuttal is necessary if we wish to improve PSPC _into_ a reasonable generative model.
> > >
> > > It is clear that the distributional metrics mentioned by the reviewer should be optimized if we wish to improve the sample quality of PSPC. However, we note that these metrics are challenging to optimize as they give no feedback as to when or how individual sampling trajectories deviate from network baselines. For this reason, we believe reporting MSE over the entire reverse process is a more useful metric as it characterizes exactly where these deviations occur. It is our view that this characterization is an important first step in any future work to improve PSPC into a useful generative model.
> > >
> > > ### **2. Traditional Patch Denoisers**
> > >
> > > We have included BM3d as a baseline in our response to the reviewers third point. However, we’d also like to clarify two differences between diffusion denoisers and the classical denoisers we reference in section 7
> > >
> > > 1. Classical denoisers are interested in sampling an image $\\mathbf{x}$ given a noisy $\\mathbf{z}$. That is, their aim is $\\mathbf{x} \\sim p_t(\\mathbf{x} | \\mathbf{z})$. By contrast, diffusion denoisers must estimate the posterior mean $\\mathbb{E}[\mathbf{x} | \mathbf{z}, t]$. While these objectives are similar for sufficiently low $t$, at intermediate noise levels and above the problems are distinct. We would not expect classical denoisers to produce reasonable posterior mean estimates for large values of $t$.
> > > 2. The noise levels on which diffusion models are trained are generally much higher than classical methods. For example, in “Field of Experts” (Roth & Black 2005), they evaluate denoising up to a maximum PSNR of 8.13, corresponding to approximately $t=0.8$ in our work.
> > >
> > > These challenges are demonstrated when using BM3d on images with higher amounts of noise. While performance for small $t$ is good, BM3d’s performance suffers beyond $t=0.1$[[Link](https://drive.google.com/file/d/1YtqRcjpzE4cuWKrtvRMyw8aMwx1HuKBf)].
> > >
> > > We will include our elaboration on the differences between traditional and diffusion denoising problems in section 7 of any camera ready revision.
> > >
> > > ### **3. Sample Visualizations**
> > > While the Gaussian denoiser does not explicitly utilize local operations to the same degree as PSPC, it is worth noting that their denoiser is primarily built upon the covariance matrix of the training dataset which captures local correlations in the data. We suspect that this is the primary reason that our denoisers have similar behaviour. Although more thorough investigation is required to test that hypothesis, we believe that it is an exciting avenue for future research.
> > >
> > > In response to the reviewer's request, we have compiled samples from a number of denoisers [[Link](https://drive.google.com/file/d/11oKbig7zhr6r-zKwjXNGHmrpdDdZVm8-)].
> > >
> > > Examining the figure, Gaussian samples do share a remarkable structural similarity to those of PSPC, but with higher saturation. Looking at the samples of closed-form diffusion models, the samples are high quality, but this is because they are all exact training set copies. Notably however, they are not always the same images as those produced by the empirical denoiser. Finally, the quality of the BM3d samples demonstrates that it is unsuitable as a diffusion denoiser.
> > >
> > > We will include this figure in the supplementary of any camera ready revision.

---

### Official Review · Reviewer_HwKw · 2025-03-12

**Overall Recommendation:** 4

**Summary:**

This paper attempts to explain the mechanism behind the ability of diffusion models to generalize beyond the training data. It starts by pointing out that this ability is due to the neural denoiser deviating from the optimal empirical denoiser (the optimal denoiser for the training set). It then shows that the function learned by neural denoisers is more similar to local empirical denoisers, namely ones which operate on small-sized patches rather than on the whole image. The paper therefore concludes that it is this tendency of neural denoisers to learn local operations that enables diffusion models to generalize well.

## update after rebuttal
The rebuttal has answered my questions. I keep my original score.

**Claims And Evidence:**

The claims are supported by clear and convincing evidence.

**Essential References Not Discussed:**

The paper draws connections to classical patch-based image restoration methods. But there were also quite a few patch-based image generation methods. Most of them in the context of learning from a single image. Starting from the classical texture-generation paper:
- J. De Bonet, "Multiresolution sampling procedure for analysis and synthesis of texture images", SIGGRAPH'97.

To more recent papers, like:
- N. Granot, et al. "Drop the GAN: In defense of patches nearest neighbors as single image generative models", CVPR'22.

These are also related to methods that learn patch statistics using GANs (e.g. SinGAN) or using diffusion models (e.g. SinFusion, SinDiffusion, SinDDM).

**Experimental Designs Or Analyses:**

I checked the experimental settings for all the experiments. In general, all experiments seem valid to me.

There are, however, several missing details that I think are important to mention and discuss:
- What was the dataset of patches used to construct each empirical patch denoiser (both for PSPC-patch and for PSPC-square)? For example, when constructing the denoised patch at location (i,j), does the computation involve all the overlapping patches extracted from all the training images, or only the patches at location (i,j) extracted from all the training images (namely, only one patch per training image)?
- Were denoised patches constructed only for the patches that are fully contained within the 64x64 image, or did patches at the boundaries also contain zero-padded regions?

These points are important for understanding how the sampling process with PSPC manages to generate sky at the top, grass at the bottom, etc. If when denoising each patch, the computation involves patches extracted from all spatial locations within the training images,  and there are no cues from padding, then the statistics for each patch is the same. Namely, patches at the upper part of the generated image should not necessarily favor sky, and patches at the bottom should not necessarily favor grass. All patches are obtained using the same operation applied to the input.

It would be great if the authors can clarify those points.

**Methods And Evaluation Criteria:**

The methods and evaluation criteria make sense.

**Other Comments Or Suggestions:**

Typo: It seems that in Eq. (8), the input to $D\theta$ should be $z$ rather than $x$.

**Other Strengths And Weaknesses:**

As mentioned above, this paper's main strength is that it provides convincing experiments that point to a plausible explanation for generalization in diffusion models.

A weakness is that the paper does not explain, and does not provide supporting experiments, how local processing induces coherent global structure. If when denoising a patch, the denoiser doesn't know from where in the image that patch was extracted, how does the denoiser know whether this patch is more likely to contain sky (if it was extracted from the upper part of the image) or grass (if it was extracted from the lower part of the image)?

**Questions For Authors:**

See comments in the sections above.

**Relation To Broader Scientific Literature:**

Understand generalization in diffusion models is a topic of vast interest, both theoretically, and with empirically.

This paper presents convincing evidence that at least part of the generalization capabilities of diffusion models is associated with the tendency of neural denoisers to learn local processing. Interestingly, the tendency to learn local processing is not a feature of a particular architecture. Even DiTs, whose architecture doesn't promote this implicit bias, learn local processing. These observations are certainly interesting to the community and may draw follow-up works that attempt to explain this tendency.

**Theoretical Claims:**

The paper doesn't present any formal theoretical claims in the forms of theorems or lemmas.

---

> ### Author Rebuttal · Authors · 2025-03-31
>
> ### **General Comment**
>
> We'd like to thank the reviewer for taking the time to read and review our paper. Furthermore, we appreciate the additional references that the reviewer has brought to our attention. Below, we have responded to several items of your review. If there are items which you feel have not been sufficiently addressed, we would welcome additional discussion on these matters.
>
> ### **Experimental Designs or Analyses:**
>
> >There are, however, several missing details that I think are important to mention and discuss:
>
> We have clarified both of the details below. Although we believe that our definitions of $C_s$ on line 199, $C_{G(t, \lambda)}$ on line 316, as well as equations 9,10 and 11 address these details, we will update the text in the camera ready revision to improve clarity on these items
>
> > - What was the dataset of patches used to construct each empirical patch denoiser (both for PSPC-patch and for PSPC-square)? For example, when constructing the denoised patch at location (i,j), does the computation involve all the overlapping patches extracted from all the training images, or only the patches at location (i,j) extracted from all the training images (namely, only one patch per training image)?
>
> For both PSPC-Flex and PSPC-Square, we construct an empirical patch set for each patch posterior mean. In the case of PSPC-Flex, this corresponds to one patch set for each pixel in the input image (ie $32^2$ patch sets for CIFAR-10 and $64^2$ patch sets for AFHQ & FFHQ). For PSPC-Square, we create a dataset for each possible square patch location with no padding and a stride of one. The total number of patch sets created for PSPC-Square varies depending on patch size. In general, we create $(h - s + 1)^2$ patch sets, where $h$ is the height of the image in pixels and $s$ is the height of the square patch in pixels. For both PSPC-Flex and PSPC-Square, each patch set is created by applying the same cropping matrix to each image in the training set. This results in patch sets with one spatially localized patch per training set image.
>
> > - Were denoised patches constructed only for the patches that are fully contained within the 64x64 image, or did patches at the boundaries also contain zero-padded regions?
>
> We did not use padding in our computations.
>
> ### **Essential References Not Discussed:**
> >The paper draws connections to classical patch-based image restoration methods. But there were also quite a few patch-based image generation methods. Most of them in the context of learning from a single image. Starting from the classical texture-generation paper:
> > - J. De Bonet, "Multiresolution sampling procedure for analysis and synthesis of texture images", SIGGRAPH'97.
> >
> >To more recent papers, like:
> > - N. Granot, et al. "Drop the GAN: In defense of patches nearest neighbors as single image generative models", CVPR'22.
> >
> >These are also related to methods that learn patch statistics using GANs (e.g. SinGAN) or using diffusion models (e.g. SinFusion, SinDiffusion, SinDDM).
>
> We thank the reviewer for drawing our attention to these highly relevant references. We will include the references mentioned in the camera ready version of the paper.
>
> ### **Other Strengths And Weaknesses:**
> >A weakness is that the paper does not explain, and does not provide supporting experiments, how local processing induces coherent global structure. If when denoising a patch, the denoiser doesn't know from where in the image that patch was extracted, how does the denoiser know whether this patch is more likely to contain sky (if it was extracted from the upper part of the image) or grass (if it was extracted from the lower part of the image)?
>
> As mentioned in the experimental design, we use spatially consistent patches for each patch posterior mean. This means that for example, when denoising patches in the upper portion of the image, the patch dataset contains patches which are more likely to contain sky than grass. We found this detail to be especially important for localized datasets such as FFHQ where facial features (noses, eyes, mouths etc.) are located in specific spatial regions of the image.
>
> Another key finding of our work is that the ideal patch size and therefore the degree of locality is anti-correlated with the level of noise. When generating a sample with PSPC-flex or PSPC-square, we initially denoise using large patches before moving to smaller patches. The larger patches therefore condition the posterior means of the later, smaller patch denoising operations. We believe this large to small patch hierarchy is important for ensuring coherent global structure
>
> ### **Other Comments Or Suggestions:**
> >Typo: It seems that in Eq. (8), the input to $D_\\theta$ should be $\\mathbf{z}$ rather than $\\mathbf{x}$
>
> Thank you for this observation. You are correct, the input to $D_\\theta$ is $\\mathbf{z}$. We will correct this mistake in the camera ready.

---

> > ### Comment · Reviewer_HwKw · 2025-04-04
> >
> > I thank the reviewers for the detailed answers. These have addressed my concerns, and would be best if clarified and discussed in the paper. I keep my original rating.

---

### Official Review · Reviewer_Se8t · 2025-03-14

**Overall Recommendation:** 5

**Summary:**

That real diffusion models generalize their training data, rather than memorize it, is not obvious: the optimal solution to a typical denoising score matching objective is the score of the (empirical) data distribution, which can only reproduce training examples. Why do diffusion models generalize, and what inductive biases determine how they generalize? The authors of this paper propose that denoising happens locally, in 'patches', rather than globally, and that this is at least partly responsible for generalization. They validate this hypothesis through a variety of numerical experiments, which show that their patch denoiser looks more like real denoisers than the 'optimal' one.

**Claims And Evidence:**

The paper has a clear hypothesis and collects an impressive set of empirical results to validate it.

**Essential References Not Discussed:**

No references come to mind.

**Experimental Designs Or Analyses:**

The experiments the authors run are well-designed and appear sound.

**Methods And Evaluation Criteria:**

The methods by which the authors test their hypothesis are reasonable, and their evaluation criteria (a mix of comparing denoisers and looking at generated samples) make sense.

**Other Comments Or Suggestions:**

line ~158: "plot four such heatmaps in Figure 3" needs a period

line ~359: "can be found in Appendix D" needs a period

**Other Strengths And Weaknesses:**

This paper is well-written, clear, and has well-made figures. It was a joy to read.

As a very minor point, it could be helpful to include some additional discussion of what these findings may mean. This inductive bias ('locality', to use Kamb and Ganguli's terminology) seems to be true for at least U-net- and vision-transformer-based diffusion models trained on images. What about other kinds of architectures or data sets?

**Questions For Authors:**

What kind of architectures does one expect to exhibit this inductive bias? If one trained a fully-connected MLP-like architecture to denoise images, would it exhibit this bias?

**Relation To Broader Scientific Literature:**

This paper makes a major contribution to the study of diffusion model generalization/memorization, which is itself a fairly major subfield of diffusion models. It also contributes meaningfully to literature on how generative models generalize and produce 'creative' output. Their main claim is kind of similar to that of Kamb and Ganguli, which they mention, but the details of their approach are a bit different, and it is likely that the two works developed somewhat independently.

**Theoretical Claims:**

The authors make no major new theoretical claims.

---

> ### Author Rebuttal · Authors · 2025-03-31
>
> ### **General Comment**
> We'd like to thank the reviewer for their thoughtful consideration of our work. Below, we have responded to specific components of your review. If you have additional questions, we would welcome further discussion.
>
> ### **Relation To Broader Scientific Literature:**
> >This paper makes a major contribution to the study of diffusion model generalization/memorization, which is itself a fairly major subfield of diffusion models. It also contributes meaningfully to literature on how generative models generalize and produce 'creative' output. Their main claim is kind of similar to that of Kamb and Ganguli, which they mention, but the details of their approach are a bit different, and it is likely that the two works developed somewhat independently.
>
> Although our method shares many similarities to Kamb & Ganguli, our work is entirely concurrent and independent. There are some differences between our methodologies and experimental results which we will highlight here, but plan to elaborate on in our camera ready revision.
>
> The primary difference between our works is that Kamb & Ganguli utilize shared patch distributions whereas we utilize patch distributions which are localized to specific image locations. We found that this localization was essential for FFHQ & AFHQ where the subject is centered within the image. In addition, Kamb & Ganguli utilize square patches exclusively, while PSPC-Flex uses localized, adaptive patch shapes and sizes. Experimentally, while Kamb & Ganguli restrict their analysis to only convolutional networks, we find similar generalization patterns exist even in convolution free architectures such as DiT and UViT.
>
> ### **Other Strengths And Weaknesses:**
> >As a very minor point, it could be helpful to include some additional discussion of what these findings may mean. This inductive bias ('locality', to use Kamb and Ganguli's terminology) seems to be true for at least U-net- and vision-transformer-based diffusion models trained on images. What about other kinds of architectures or data sets?
>
> Please see our response to this point in the Questions for Authors section
>
> ### **Other Comments Or Suggestions:**
> >line ~158: "plot four such heatmaps in Figure 3" needs a period
> >
> >line ~359: "can be found in Appendix D" needs a period
>
> We'd like to thank the reviewer for identifying these items. We will fix both in the camera ready revision.
>
> ### **Questions For Authors:**
> > What kind of architectures does one expect to exhibit this inductive bias? If one trained a fully-connected MLP-like architecture to denoise images, would it exhibit this bias?
>
> This is an interesting open question in our opinion. Although more rigorous study is required, we would guess that the inductive biases are probably due to interplay between the diffusion process and the correlations in the data. For example, in datasets where the features are not locally correlated, (ie shuffling the data dimensions), we would not expect this local inductive bias to be useful.
>
> Our observations in this paper seem to suggest that network architectures do not significantly affect the generalization behaviour. We would therefore expect MLPs to have similar properties to DiT & U-Net architectures when trained on the same data. This hypothesis is somewhat supported by prior art. For example, Li et al. (2024) show that the optimal linear denoiser is the Gaussian denoiser which produces outputs which are quite similar to our method.

---

### Decision · Program_Chairs · 2025-05-01

**Decision:**

Accept (spotlight poster)

**Comment:**

This paper advances our understanding of why diffusion models do not simply memorize training data but instead exhibit genuine generalization. Specifically, the authors propose that diffusion denoisers exploit a “local inductive bias,” whereby outputs at each spatial position in an image are primarily shaped by nearby pixels. This hypothesis is intriguing, and the evidence the authors provide is convincing. The consensus among the majority is that the paper’s novel insights, careful empirical investigation, and thorough rebuttal responses justify acceptance.

One reviewer, however, remains unconvinced, expressing a desire for more rigorous theoretical grounding and more extensive tests to confirm that “locality” truly drives the observed generalization.